# Direct regulation of the voltage sensor of HCN channels by membrane lipid compartmentalization

Lucas J. Handlin[1] & Gucan Dai [1] ✉

Ion channels function within a membrane environment characterized by dynamic lipid compartmentalization. Limited knowledge exists regarding the response of voltage-gated ion channels to transmembrane potential within distinct membrane compartments. By leveraging fluorescence lifetime imaging microscopy (FLIM) and Förster resonance energy transfer (FRET), we visualized the localization of hyperpolarization-activated cyclic nucleotide-gated (HCN) channels in membrane domains. HCN4 exhibits a greater propensity for incorporation into ordered lipid domains compared to HCN1. To investigate the conformational changes of the S4 helix voltage sensor of HCN channels, we used dual stop-codon suppression to incorporate different noncanonical amino acids, orthogonal click chemistry for site-specific fluorescence labeling, and transition metal FLIM-FRET. Remarkably, altered FRET levels were observed between VSD sites within HCN channels upon disruption of membrane domains. We propose that the voltage-sensor rearrangements, directly influenced by membrane lipid domains, can explain the heightened activity of pacemaker HCN channels when localized in cholesterol-poor, disordered lipid domains, leading to membrane hyperexcitability and diseases.

Voltage-gated ion channels (VGICs) play a crucial role in various physiological functions, such as the initiation and termination of action potentials, neurotransmitter release, sensing pain signals, and processing cognitive activities[1]. Most of VGICs possess conserved voltage-sensing domains (VSD) that are characterized by containing positively charged arginines[1]. These arginines are positioned across the transmembrane electric field and directly sense changes in membrane voltage[1]. Among all the VGICs, the hyperpolarization-activated cyclic nucleotide-gated (HCN) channels are unique because they are activated by membrane hyperpolarization and deactivated by depolarization[2]. This voltage-dependent gating behavior of HCN channels is opposite to other types of VGICs, which are activated by membrane depolarization and deactivated by hyperpolarization[1,3]. The activation of HCN channels during hyperpolarization enables inward sodium currents during the final phase of an action potential. This helps bring the membrane potential close to the threshold of firing the next action potential. Consequently, HCN channels play critical roles in regulating the autonomic rhythmicity of excitable cells.

HCN channels are recognized for their crucial roles in cardiac pacemaking, pain sensation, and epileptic seizures[4–7]. In humans, there are four HCN channel isoforms, encoded by the HCN1–4 genes. HCN1 is the primary isoform expressed in neurons, while HCN4 is the primary isoform expressed in cardiac pacemakers. HCN2 has considerable expression in both neurons and the heart, while HCN3 has comparatively low expression levels. HCN2 and HCN4, like their function as pacemakers in the sinoatrial node of the heart, also regulate the excitability of primary nociceptor neurons in the dorsal root ganglion[8,9]. For most HCN channels, especially HCN2 and HCN4, cyclic adenosine monophosphate (cAMP) is an agonist that promotes hyperpolarization-dependent channel opening[10,11]. This cAMP-mediated modulation of HCN channels underlies the sympathetic regulation of heartbeat[12,13].

[1]Edward A. Doisy Department of Biochemistry and Molecular Biology, Saint Louis University School of Medicine, 1100 South Grand Blvd., St. Louis, MO 63104, USA. ✉e-mail: gucan.dai@health.slu.edu

The structural mechanism underlying the unique hyperpolarization-dependent activation of HCN channels was revealed using fluorescence imaging, computational modeling, and high-resolution cryo-electron microscopy[2,14–18]. HCN channels share a common architecture with other VGICs, including cyclic nucleotide-gated (CNG) channels and KCNH voltage-gated potassium channels. Four similar or identical subunits surround a centrally located pore, with each subunit comprising a voltage-sensing domain (VSD) consisting of four transmembrane helices (S1–S4), a pore domain (PD) consisting of S5 and S6 transmembrane helices, a selectivity filter, and a C-terminal C-linker and cyclic nucleotide-binding domain (CNBD)[19]. Upon hyperpolarization, the voltage sensor S4 helix, where the positively charged arginines are situated, moves towards the intracellular side nearly 10 Å in distance, exhibiting a bending motion that further tilts the S5 helix[2]. Due to an unusual "non-domain-swapped" subunit configuration, the S5 helix movement causes the opening of the channel pore[16]. One notable feature of HCN channels is that the S4 helix is exceptionally long, nearly twice as that of other VGICs[14,20]. This feature suggests that a "hydrophobic mismatch"[21] could occur when HCN channels reside in thinner lipid bilayers, providing an unconventional way to regulate voltage sensing and channel gating.

The cell membrane lipid composition is heterogeneous and contains distinct lipid compartments (domains) for localized cell signaling[22]. These compartments include liquid-ordered ($L_o$) membrane fractions that contain more cholesterols and sphingolipids than liquid-disordered ($L_d$) domains[22–26]. $L_o$ domains, which are thicker than $L_d$ domains, can range in size from <10 to >300 nm, and can coalesce or divide dynamically[27,28]. These ordered domains have drawn growing attention due to their involvement in various processes, such as Alzheimer's disease[29–32], viral infection of cells[33,34], inflammation[35,36] and neuropathic pain[37], and have been visualized using super-resolution microscopy, single-molecule fluorescence, and cryo-electron tomography[38,39]. Furthermore, the cardiac HCN4 channel opening and its voltage sensitivity are potentiated by the disruption of $L_o$ domains through the use of the cholesterol-depleting reagent β-cyclodextrin (β-CD)[40,41]. This potentiation results in an increased action-potential frequency of cardiac pacemaker cells by facilitating HCN4 gating[40]. In contrast, the steady-state activation of human HCN1 channels is not affected by β-CD, suggesting that the gating of neuronal HCN1 is resistant to the disruption of $L_o$ domains[41].

This study employed various fluorescence techniques, including FLIM-FRET, to investigate the membrane localization of human HCN channels. The results reveal that the voltage-sensing domain appears to undergo rearrangement when present in different membrane compartments. This rearrangement can either facilitate or hinder the hyperpolarization-dependent movement of the voltage sensor required for channel opening. Our results strongly suggest that the rearrangement of voltage-sensing helices of VGIC can be directly regulated by changes in the order and thickness of the lipid bilayer. These findings can explain the distinct response of HCN isoforms to the disruption of lipid domains. This mechanism could account for the hyperexcitability observed in sensory neurons or cardiac pacemakers under disease conditions when they lose ordered membrane domains or cholesterols.

## Results

### Localization of HCN channels in different membrane domains

Traditional methods used to investigate the localization of HCN channels in lipid domains have involved detergent solubilization, followed by fractional isolation of membrane lipid components. In our study, we adopted a different approach by using two lipidated peptides fused with fluorescent proteins to probe the localization of HCN channels in live cells[42,43]. The L10 probe, like our earlier work, is derived from the first 10 amino acids located at the amino terminus of Lck kinase[42,43]. These 10 amino acids encompass two palmitoylation sites

that facilitate localization to the ordered lipid domains. In contrast, the S15 probe is derived from the first 15 amino acids at the amino terminus of Src kinase. It comprises one myristoylation site that assists in localizing to disordered domains (Fig. 1a). The L10 and S15 probes have a cyan fluorescent protein (CFP) fused to their carboxyl-terminal end, which was used as a Förster resonance energy transfer (FRET) donor with the human HCN constructs that contain the FRET acceptor yellow fluorescent protein (YFP). Human HCN1 and HCN4 channels were compared in our study, because these two isoforms are the primary isoforms present in neurons and cardiac pacemakers, respectively. They exhibit distinct gating behaviors (Supplementary Fig. 1); human HCN1 channels (hHCN1) open much faster and show higher sensitivity to voltage than human HCN4 channels (hHCN4). Notably, hHCN4 is significantly potentiated by lipid domain disruption, whereas hHCN1 remains unaffected by this disruption[40,41] (also see Supplementary Fig. 1f, g).

FRET is a nonradiative phenomenon that involves the transfer of energy between donor and acceptor molecules. The efficiency of FRET is directly influenced by the proximity of donors and acceptors. This characteristic makes FRET a highly sensitive spectroscopic tool for measuring distances, particularly valuable in measuring the localization of membrane proteins within small lipid domains. In our study, we employed two distinct approaches to quantify FRET between the lipid domain probes L10-CFP or S15-CFP and the HCN-YFP channels. The first method involved analyzing the fluorescence spectra (referred to as spectral FRET), while the second method relied on measuring fluorescence lifetime.

Spectral FRET was determined by calculating the emission spectra of CFP and YFP using a microscope-attached spectrograph. This approach directly measures the degree of sensitized emission of the FRET acceptor upon excitation of the FRET donor. Corrections were applied for bleed-through and crosstalk caused by the spectral properties of the fluorophores (see the "Methods" section)[43]. The FRET spectrum was obtained by measuring the fluorescence spectrum when both L10 or S15-CFP and HCN1 or HCN4-YFP channels were present. Template emission spectra from tsA-201 cells expressing CFP probes alone were used to correct for bleed-through of CFP emission. The bleed-through was corrected by scaling and subtracting the CFP-only spectrum. The resulting $F_{FRET}$ spectrum was then divided by the YFP emission spectrum directly excited by YFP excitation to generate a fraction of the sensitized emission of YFP by only exciting CFP. The apparent FRET efficiency was calculated and plotted against the ratio of CFP and YFP fluorescence intensities (see the "Methods" section). By fitting with a previously established equation (Eq. (5))[43], the true FRET efficiency ($E_{max}$) at high $F(CFP)/F(YFP)$ ratios was estimated, providing information about the distance between the L10 or S15 probes and HCN channels. In our overexpression system in tsA cells, we observed higher FRET efficiency ($E_{max} = 12.6\%$) between S15-CFP and hHCN1-YFP compared to L10-CFP and hHCN1-YFP ($E_{max} = 4.3\%$), indicating that the majority of hHCN1 channels reside in disordered $L_d$ membrane domains. Conversely, the FRET values were similar for the L10-CFP/hHCN4-YFP ($E_{max} = 15.5\%$) and S15-CFP/hHCN4-YFP ($E_{max} = 13.8\%$) pairs, suggesting that a significant number of hHCN4 channels colocalize with ordered $L_o$ membrane domains (Fig. 1d).

To provide further evidence supporting the preferential localization of cardiac HCN4 channels in $L_o$ domains compared to neuronal HCN1 channels, we employed fluorescence lifetime imaging microscopy (FLIM). FLIM allows the measurement of FRET by quantifying the reduction in donor lifetime upon the addition of the FRET acceptor. The advantage of FLIM is that it minimizes errors arising from non-specific background fluorescence and photobleaching, as fluorescence lifetime is a more distinct and precise characteristic of a fluorophore compared to its emission or absorption spectrum. Combined with our laser-scanning confocal microscope, we were able to

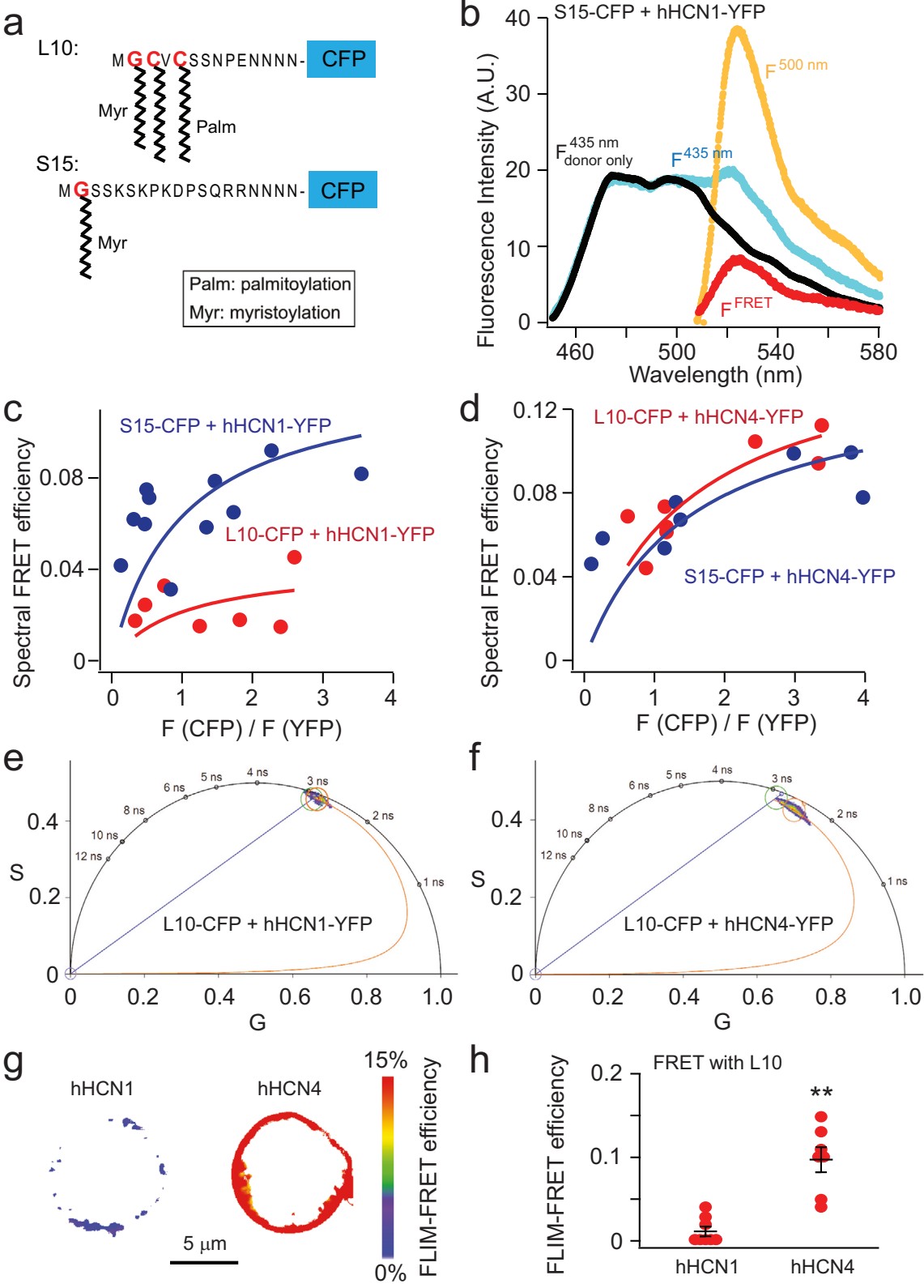

acquire more spatial information in live-cell FLIM experiments, enabling higher-throughput data collection.

We employed the phasor approach of frequency domain FLIM[44] to measure the lifetime of L10-CFP in the presence or absence of coexpressed hHCN-YFP channels in tsA cells. Through discrete Fourier transform, the lifetime data at each image pixel were represented as points on a Cartesian coordinate called a phasor plot, characterized by

a universal semicircle (Supplementary Fig. 2)[44–46]. Pure single-exponential lifetimes aligned with the semicircle, while multi-exponential lifetimes appeared inside the semicircle. Shorter lifetimes exhibited clockwise movement along the semicircle, indicating increased FRET. The phasor approach offers several advantages[44]: (1) it is model-independent, providing fast and unbiased quantification of fluorescence lifetimes without the need for exponential fitting; (2) it

**Fig. 1 | Using lipidated fluorescent probes and FLIM-FRET to quantify the lipid-domain localization of human HCN channels. a** Schematic illustration of L10- and S15-CFP probes. **b** Representative emission spectra used to calculate FRET between S15-CFP and hHCN1-YFP. Cells that expressed CFP-containing probes alone were used to generate a template CFP emission spectrum ($F_{donor\ only}^{435\ nm}$ excited by 435 nm light in black) to correct for the bleed-through of the CFP emission in the wavelength range for YFP emission. To determine the FRET spectrum, the fluorescence spectrum was measured when both CFP and YFP were present ($F^{435\ nm}$ spectrum in cyan). The $F_{donor\ only}^{435\ nm}$ spectrum in **b** was then scaled and subtracted from the $F^{435\ nm}$ spectrum, yielding the resulting $F^{FRET}$ spectrum (in red). Then, the peak intensity of the $F^{FRET}$ spectrum was divided by the peak intensity of the YFP

emission spectrum directly excited by the YFP excitation of 500 nm light ($F^{500\ nm}$ in yellow). A.U. arbitrary units. **c** and **d** Relationship between the apparent FRET efficiency and the ratio of peak intensities of CFP versus YFP for FRET between L10-CFP/S15-CFP and hHCN1-YFP/hHCN4-YFP. **e** and **f** Phasor plots of FRET pairs of L10-CFP with hHCN1-YFP (left) and L10-CFP with hHCN4-YFP (right), using the scale of 39.5 MHz modulation frequency. **g** Exemplar FLIM-FRET heatmap images of cultured tsA cells, highlighting the low FRET between L10 and hHCN1, and the high FRET between L10 and hHCN4 channels. **h** Summary of the FLIM-FRET efficiency measured between L10 and HCN isoforms. Data shown are mean ± s.e.m., $n = 8$ cells for hHCN1, and $n = 7$ cells for hHCN4. Two-sided Student's $t$-test was used as the statistical analysis method; ** $p = 8.7E{-}05$.

allows for multiple-cursor mapping to identify distinct lifetime species in confocal images; (3) it enables linear decomposition of complex lifetime species into individual components, facilitating the identification of different FRET states in both ensemble and single-molecule experiments.

We mapped the lifetimes of L10-CFP alone using a green cursor on the phasor plot (Fig. 1e and f). In the presence of hHCN4-YFP, the CFP lifetime, represented by the red cursor, exhibited clockwise movement towards shorter lifetimes. In contrast, the change in CFP lifetime was less pronounced in the presence of hHCN1-YFP. FRET efficiency was calculated using the FRET trajectory method in the phasor plot[44]. The FRET trajectory started from the zero FRET efficiency, overlapping with the donor-alone cursor, and curved through the measured phasor species, ending at the theoretical 100% FRET efficiency at the origin (0, 0) coordinate where the background noise is mainly contributed from the white noise of infinite lifetime (Fig. 1e and f). Furthermore, FRET efficiency at each pixel was determined using the FRET trajectory function, generating a FRET efficiency-based heatmap for a single cell (Fig. 1g). The averaged FRET efficiencies of these FRET pairs are summarized in Fig. 1h, and they fall within a similar range as the FRET efficiency calculated using spectral FRET. These findings align with the spectral FRET data, supporting the localization of hHCN4 channels in $L_o$ domains to a greater extent than hHCN1 channels.

## Probing the lipid-domain localization of HCN channels using cholera toxin subunit B

To overcome the limitations associated with YFP fusion and probe lipidation, we developed an alternative FRET method to further define the isoform-specific localization of HCN channels in different lipid domains. This method uses cholera toxin subunit B (CTxB), a pentameric subunit derived from *Vibrio cholerae*, the bacterium responsible for causing cholera. CTxB has a high affinity for GM1 gangliosides, which are known to localize to $L_o$ domains and serve as a marker for ordered membrane domains[24,26]. Importantly, CTxB binding occurs extracellularly and provides specific information about the localization of $L_o$ domains, despite potentially inducing some coalescence effects due to its multivalent interaction with GM1.

To specifically label the transmembrane regions of HCN channels, we employed the amber stop-codon suppression technique, which expands the genetic code to incorporate a noncanonical amino acid (ncAA) into the VSD of hHCN1 and hHCN4 channels (Fig. 2a). This approach uses an orthogonal transfer RNA (tRNA)-aminoacyl tRNA synthetase pair, where the tRNA possesses an anticodon that matches the amber stop codon. When the ribosome encounters the amber stop codon during translation, the orthogonal ncAA is selectively incorporated into the protein sequence while translation continues[47–50]. In our study, we used the archaea pyrrolysine (Pyl)-based amber stop-codon suppression system, enabling the incorporation of trans-cyclooct-2-ene-L-lysine (TCO*K) into the channels (Fig. 2b)[49]. TCO*K can be conjugated to a tetrazine-linked dye using the strain-promoted inverse electron-demand Diels–Alder (IEDDA) cycloaddition reaction, which is known for its fast reaction rates[49,51]. This choice of the Pyl system also allows for dual-codon suppression in mammalian cells, as described

later in the paper. Furthermore, tetrazine-conjugated dyes, such as tetrazine-Alexa Fluor 488 (AF-488), exhibit a significant fluorescence increase upon IEDDA reaction. To confirm the success of the reaction, we measured the fluorescence lifetime of tetrazine-AF-488 using phasor FLIM and observed a substantial increase in lifetime upon the addition of TCO*K (Fig. 2c), indicating the completion of the IEDDA reaction.

To investigate the incorporation of TCO*K into hHCN channels, we used a hHCN4-L374TAG-mCherry construct. The residue L374 is positioned at the amino-terminal end of the S4 transmembrane helix, which places it in proximity to the extracellular surface of the plasma membrane, allowing accessibility for labeling with tetrazine-conjugated dyes. The mCherry fluorescence served as an indicator of full-length hHCN4 channel expression. When the hHCN4-L374TAG-mCherry construct and TCO*K amino acids were present but the aminoacyl tRNA synthetase (PylRS-Y125A from Methanogenic archaeon ISO4-G1) was absent, minimal mCherry fluorescence was observed following transfection. However, the addition of the synthetase rescued channel expression, resulting in strong mCherry fluorescence with membrane-localized signals (Fig. 2d). As AF-488 is sensitive to its environment, the lifetime of tetrazine AF-488 differed from that of freely diffusing TCO*K/tetrazine AF-488 after labeling cells expressing hHCN4-L374TCO*K (Fig. 2c). The fluorescence of AF-488 exhibited a positive correlation with mCherry, indicating that a significant portion of the incorporated TCO*K molecules was successfully labeled with tetrazine AF-488 (Supplementary Fig. 3).

We established a FRET pair involving the donor tetrazine AF-488, which was attached to another position within the S4 helix of HCN channels, close to the lipid membrane. The acceptor was CTxB Alexa Fluor 594 (AF-594) conjugate (Fig. 2e). The specific amber codon sites used were I382TAG for hHCN4 or the corresponding site I262TAG for hHCN1. These sites were chosen to bring the labeled AF-488 in closer proximity to the outer leaflet of the membrane compared to the L374TAG site of HCN4 and the L254TAG site of HCN1. The Förster distance ($R_0$) for the AF-488/AF-594 FRET pair is 59 Å (according to FPbase.org), and significant FRET should be observed if the channels are localized within GM1-rich $L_o$ domains. By implementing phasor plot FLIM, we observed a more pronounced clockwise shift in the AF-488 lifetime for hHCN4 channels compared to hHCN1 channels when CTxB-AF594 was added to cells transfected and labeled using the TCO*K/tetrazine AF-488 strategy. Similarly, a FLIM-FRET efficiency-based heatmap of single cells can be generated to visualize this difference in FRET (Fig. 2g). These findings suggest that HCN4 exhibits a higher level of CTxB-mediated FRET compared to HCN1, which serves as additional evidence supporting the localization of hHCN4 within $L_o$ domains in contrast to hHCN1 (Fig. 2f–h). Moreover, these results indicate that hHCN4 is more vulnerable to disturbances in lipid domain integrity.

## Rearrangement of the voltage sensor reported by the fluorescent amino acid L-Anap

During membrane hyperpolarization, movement of the S4 voltage-sensing helix in HCN channels leads to significant changes in the

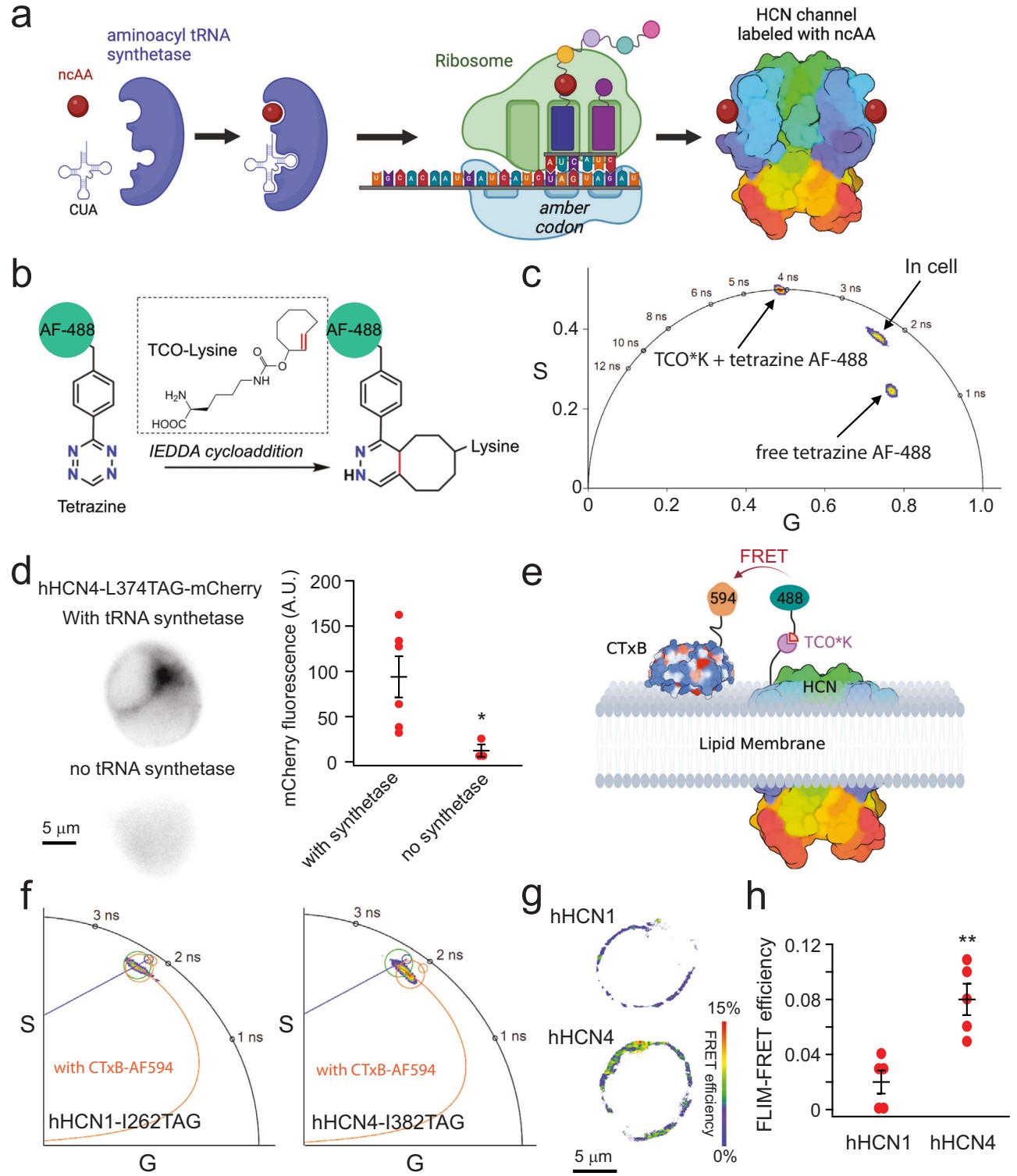

local environment of residues near the charge transfer center. In our previous studies, we demonstrated that replacing a serine residue near the hydrophobic constriction site with a fluorescent non-canonical amino acid called L-Anap[52,53] (S346Anap in sea urchin HCN channels) resulted in a substantial increase in fluorescence upon hyperpolarization[2]. This serine residue corresponds to S392 in hHCN4 and is conserved across all human HCN channels. Additionally, this specific site is in proximity to the region where the S4 helix undergoes bending during its downward movement, which is crucial for the channel opening process[2,14].

Using FLIM imaging and the amber stop codon suppression technique, we observed that the fluorescence lifetime of L-Anap incorporated into the hHCN4-S392TAG construct closely resembled the lifetime of free L-Anap in hydrophobic solvents like ethanol, but differed from its lifetime in a water-based buffer (Fig. 3b). This finding suggests that the S392 site is suitable for L-Anap incorporation and likely resides in a hydrophobic environment. When cells were exposed to a high potassium (120 mM KCl) solution, resulting in depolarization (also see Supplementary Fig 4), the lifetime of hHCN4-S392Anap decreased (Fig. 3b and c). This observation is consistent with our

**Fig. 2 | Combining ncAA incorporation, click chemistry, and FLIM-FRET to probe the localization of HCN channels in membrane domains. a** Cartoon illustration of noncanonical amino acid incorporation using amber stop-codon suppression. **b** IEDDA reaction using TCO*K and AF488-tetrazine. **c** Lifetime of tetrazine AF-488 when alone, after reacting with TCO*K, and when incorporated into HCN4 channels in the phasor plot of FLIM. **d** Conditions with/without the tRNA synthetase and the summary of the mCherry fluorescence intensity. Data shown are mean ± s.e.m., $n = 3$ cells for the condition without the synthetase and $n = 6$ cells with the synthetase. Membrane localized-mCherry fluorescence indicates the expression of full-length channels. Two-sided Student's $t$-test was used as the statistical analysis method; *$p = 0.046$. A.U. arbitrary units. **e** Cartoon illustration of the

FRET pair between a TCO*K/Tetrazine AF-488-labeled HCN channel and the CTxB-AF-594 conjugate. Only one fluorophore is shown for the tetrameric HCN channel and the pentameric CTxB. **f** Phasor plots showing the different degrees of lifetime shortening after the application of CTxB-AF-594, comparing TCO*K/Tetrazine AF-488 labeled to the hHCN1-I262 site versus to the hHCN4-I382 site. **g** Exemplar FLIM-FRET heatmap images of cultured tsA cells, highlighting the low FRET between hHCN1 and CTxB, and the high FRET between hHCN4 and CTxB, based on the experiments in the panel **f**. **h** Summary of the measured FLIM-FRET efficiency between HCN channels and CTxB. Data shown are mean ± s.e.m. $n = 5$ cells. Two-sided Student's $t$-test was used as the statistical analysis method; **$p = 2.9E{-}03$.

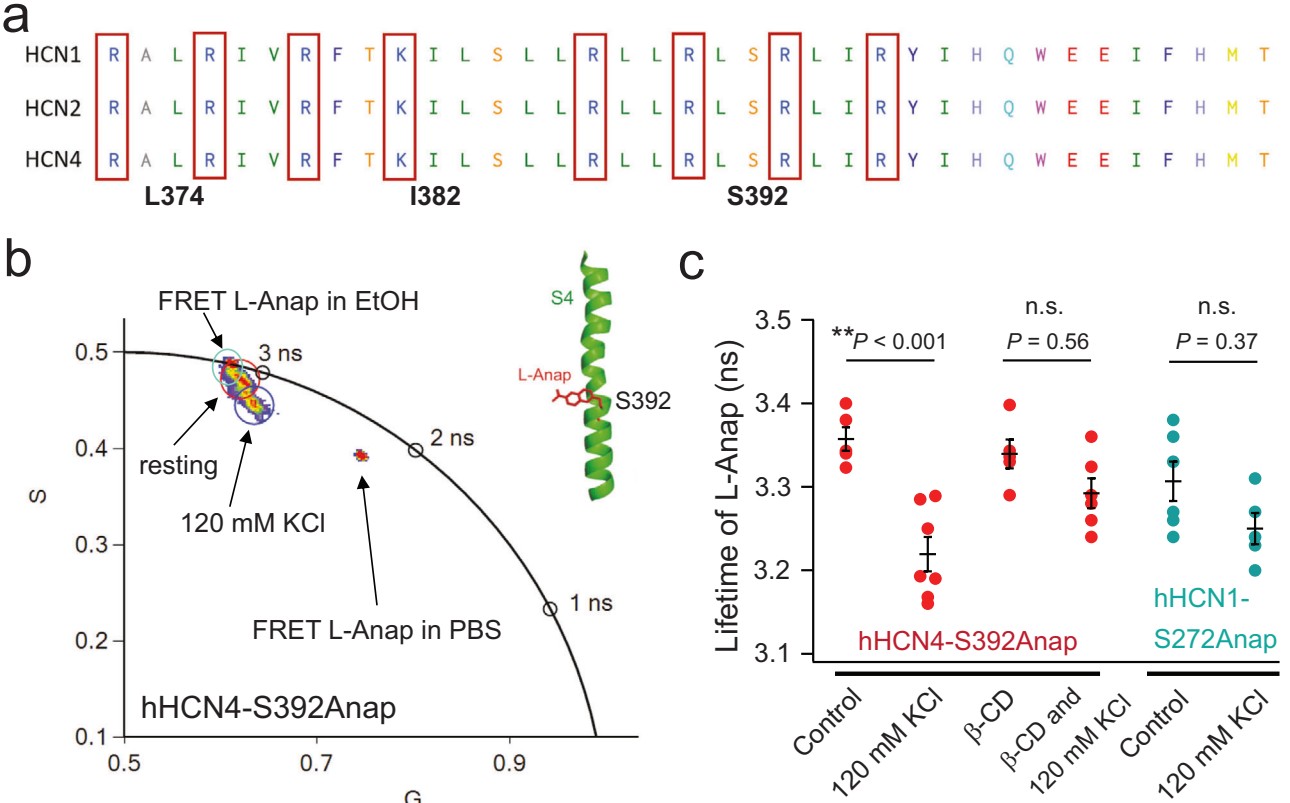

**Fig. 3 | Probing the change of the lifetime of L-Anap incorporated to the key serine residue in the voltage-sensing S4 helix of HCN channels. a** Sequence alignment of the S4 helix voltage-sensor region of human HCN1, HCN2, and HCN4 channels. The numbers correspond to specific residues of HCN4. **b** Phasor FLIM imaging of L-Anap incorporated in the middle of the S4 helix of the HCN4 channel (S392 site) expressed in tsA201 cells versus the free L-Anap in PBS buffer, and in ethanol. Application of the PBS buffer with 120 mM KCl decreased the lifetime. **c** Effects of the 120 mM KCl and/or 5 mM β-cyclodextrin on the lifetime of L-Anap for hHCN4-S392Anap ($n = 5$ cells for the control, $n = 7$ cells for the 120 mM KCl condition, $n = 5$ cells for the condition after β-CD, and $n = 6$ cells for the condition of

120 mM KCl after β-CD), as well as the effect of the 120 mM KCl on the lifetime of L-Anap for hHCN1-S272Anap ($n = 6$ cells for the control and $n = 5$ cells for the 120 mM KCl condition). Data shown are mean ± s.e.m. One-way ANOVA, followed by Tukey's post hoc tests and pairwise comparisons (no adjustment), was used for these six groups as the statistical method. Key comparisons are highlighted as $p = 3.3E{-}04$ between the control and the 120 mM KCl for hHCN4-S392Anap, $p = 0.56$ between groups with and without 120 mM KCl, and in the presence of β-CD for hHCN4-S392Anap, $p = 0.37$ between the control and the 120 mM KCl for hHCN1-S272Anap; *$p < 0.05$, **$p < 0.01$.

previous finding that depolarization reduces the quantum yield of L-Anap at the corresponding position in sea urchin HCN channels[2]. To further investigate the impact of membrane lipid domains, we tested whether extraction of cholesterols using β-cyclodextrin (β-CD), a common method for disrupting $L_o$ domains, could affect the depolarization-induced change in Anap lifetime[42]. Acute application of 5 mM β-CD did not alter the Anap lifetime under resting conditions, but significantly attenuated the change in Anap lifetime caused by the high potassium solution (Fig. 3c). Moreover, comparable experiments were performed by introducing L-Anap into the equivalent site S272TAG of hHCN1. Given that hHCN1 predominantly localizes within $L_d$ domains, the application of 120 mM KCl did not yield a notable

reduction in Anap lifetime, similar to when β-CD was applied in the context of hHCN4. These results suggest that the voltage sensor rearrangement in HCN4 may differ when the membrane lipid domain is disrupted.

## Lipid domain-dependent rearrangement of the voltage sensor of HCN channels

Conventional methods for investigating the structural mechanisms of ion channels often involve site-specific fluorescence labeling using cysteine-mediated techniques. However, this approach requires the elimination of nonspecific cysteines, which can potentially alter the characteristics of ion channels. To gain deeper insights into the

voltage-sensing domain (VSD) rearrangement of HCN channels, we employed a dual-stop codon-suppression strategy, incorporating two distinct noncanonical amino acids (ncAAs) within a single HCN subunit.

In our study, we employed a recently developed system that incorporates two orthogonal tRNA/Pyl aminoacyl tRNA synthetase (PylRS) pairs[49]. This system consisted of a G1 PylRS-Y125A variant in conjunction with an improved G1 tRNA, which contains two mutations (A41AA and C55A), along with an acceptor stem portion derived from the Pyl tRNA of *M. alvus* Mx1201, referred to as hyb*PylT[49]. Furthermore, we used tRNA (M15-PylT)/PylRS pairs from *M. mazei/M. barkeri* (Mma). Importantly, when employed in mammalian cells, the tRNA/PylRS from *M. alvus* does not cross-react with the tRNA/PylRS from *M. mazei/M. barkeri*. The hyb*PylT tRNA enables amber stop-codon suppression, encoding TCO*K, while the M15-PylT tRNA enables ochre stop-codon suppression, encoding N-propargyl-ʟ-lysine (ProK)[49]. Although this system has been successfully applied in studies involving green fluorescent protein and other membrane proteins[49], its potential application to ion channels has not been explored previously.

To fluorescently label the noncanonical amino acids (ncAAs), we employed the IEDDA reaction to conjugate TCO*K with tetrazine-conjugated AF-488, as mentioned earlier. In the case of ProK, we used copper(I)-catalyzed azide-alkyne cycloaddition (CuAAC) to attach an azide-conjugated Alexa Fluor 555 (AF-555). Compared to strain-promoted azide-alkyne cycloaddition (SPAAC), CuAAC offers significantly faster reaction rates, making it more suitable for live-cell experiments[51]. Moreover, by integrating the picolyl group into the azide dyes, the reaction rate of CuAAC is further accelerated. We employed an AF-555 picolyl azide, which facilitated rapid attachment to ProK at room temperature within a few minutes, minimizing the potential toxicity of copper to cells.

We introduced stop codons at specific sites in the voltage-sensing domain (VSD) of hHCN1 and hHCN4 channels to measure potential rearrangements of the VSD upon disruption of lipid domains using β-CD. The selected sites for the amber TAG codons were within the S4 helix, while the ochre TAA codon was positioned at the relatively stationary S1–S2 linker, both near the extracellular side of the membrane (Fig. 4c). To differentiate from the TAG and TAA sites, we engineered two consecutive opal (TGA) codons as the functional stop codons. As the stop codons were situated in the transmembrane regions, truncated proteins would not be efficiently trafficked to the membrane. Since the ochre site precedes the amber site, successful labeling of TCO*K with AF-488-tetrazine necessitated the incorporation of ProK as well. Additionally, the fluorescence intensity of AF-488 exhibited a linear correlation with that of AF-555 in individual cells, indicating that fluorescence signals predominantly originated from channels dually labeled with AF-488 and AF-555 (Fig. 4b). The fluorescence lifetime of both AF-488 and AF-555 was measured from cell membranes (Fig. 4d). Furthermore, we observed ionic currents mediated by HCN channels incorporating TCO*K and ProK (hHCN1-L254TAG/T171TAA and HCN4-L374TAG/N291TAA), which generated hyperpolarization-activated currents with a conductance–voltage relationship profile similar to wild-type channels (Supplementary Fig. 5).

We employed two FRET pairs for hHCN4, with analogous pairs used for hHCN1. The FRET pairs for hHCN4 consisted of L374TAG/N291TAA and I382TAG/N291TAA. Upon labeling the donor site with AF-488 using the IEDDA reaction, we observed donor-only lifetime species in the phasor plot. Subsequent CuAAC reaction for acceptor site labeling resulted in a clockwise shift of the donor lifetime towards shorter lifetimes for both FRET pairs of hHCN4. The shortening of the donor lifetime was more pronounced for the L374TAG/N291TAA pair compared to the I382TAG/N291TAA pair (Fig. 4e, f, and h), indicating that the I382 site is further from N291ProK, resulting in lower FRET efficiency. The averaged FRET efficiencies are summarized in Fig. 4h. As a control experiment, hHCN4-L374TAG-mCherry was unable to

incorporate ProK, allowing only AF-488 labeling of the channel. Consequently, a minimal shift in donor lifetime was observed after CuAAC, confirming that our FRET measurements are specific to ProK incorporation at the ochre site. The FRET efficiencies of HCN1 L254TAG/T171TAA and I262TAG/T171TAA were slightly lower compared to HCN4 channels. Similar to HCN4 FRET pairs, the FRET efficiency displayed site dependence, with the donor at the more amino-terminal end of the S4 helix exhibiting higher FRET. Additionally, acute application of 5 mM β-CD to deplete cholesterol substantially reduced the FRET of the two hHCN4 FRET pairs, while only moderately affecting the FRET of the L254TAG/T171TAA pair in hHCN1 (Fig. 4h). These findings imply that the conformation of the voltage-sensing domain (VSD) in HCN4 channels is influenced by membrane lipid domains. In contrast, HCN1 channels, predominantly found in disordered domains, appear to be less susceptible to changes induced by β-CD treatment.

Furthermore, considering the tetrameric assembly of HCN channels, inter-subunit FRET was anticipated based on the $R_0$ value of 68.5 Å for AF-488/AF-555 (according to FPbase.org). For instance, regarding the L254TAG/T171TAA pair in hHCN1, the β-carbon distance between adjacent subunits is 47 Å and the β-carbon distance between diagonal subunits is 77 Å, based on the cryo-EM structure of hHCN1. However, due to the presence of both inter- and intra-subunit FRET, straightforward distance estimation was hindered. To overcome this, we applied transition metal FRET, which is sensitive to a much shorter distance range than traditional FRET pairs.

## Using transition metal FRET (tmFRET) to assess the rearrangement of the HCN voltage sensor

We used tmFRET to measure short-range voltage-sensor rearrangements with a site-specifically labeled fluorophore and transition-metal acceptors to precisely monitor the movements of protein backbones[48,54–56]. In tmFRET, nonfluorescent transition metal cations, such as $Co^{2+}$ and $Cu^{2+}$, were used as FRET acceptors to efficiently quench the fluorescence of donor fluorophores in a highly distance-dependent manner[2,16,48,57]. tmFRET offers distinct advantages over traditional FRET for studying protein conformational changes. One key feature is that transition metal ions have a small extinction coefficient, resulting in a working distance range of ~10–25 Å, which aligns well with the scale of structural rearrangements in proteins. Furthermore, tmFRET exhibits reduced orientation dependence due to the isotropic nature of the metal ion, which possesses multiple transition dipole moments[2,55].

To create a tmFRET pair, we used the same dual-codon suppression strategy, but with a variation in the labeling of ProK. In this case, ProK was reacted with an azido mono-amide-1,4,7,10-tetra-azacyclododecane-1,4,7,10-tetraacetic acid (DOTA)[58] (Supplementary Fig. 6). Binding of transition metals to DOTA, similar to other transition metal-chelators containing a cyclen ring, exhibits high affinity and results in an increase in the light absorption of transition metals (Supplementary Fig. 6). The emission spectrum of AF-488 overlaps with the absorption spectra of $Co^{2+}$–DOTA and $Cu^{2+}$–DOTA complexes (Fig. 5a). The measured $R_0$ values were 15.1 Å for the AF-488/$Cu^{2+}$–DOTA pair and 13.9 Å for the AF-488/$Co^{2+}$–DOTA pair (Fig. 5b). Since the linker of DOTA is shorter than that of the picolyl-azide AF dyes, the position of the transition metal is closer to the protein backbone as well as to the tmFRET donor AF-488 (Fig. 5a). Due to its short $R_0$, intersubunit FRET is expected to be negligible. As a result, tmFRET predominantly occurs within the same subunit, allowing for more accurate reporting of voltage-sensor rearrangements compared to the AF-488/AF-555 pair.

Phasor FLIM was used to measure the degree of quenching of the tmFRET donor after applying the transition metal acceptor. Incubating the cells with $Cu^{2+}$-DOTA to achieve the CuAAC substantially shortened the AF-488 lifetime, reflecting a high FRET and close distance between the FRET pairs (Fig. 5c and g). The nonspecific quenching efficiency of

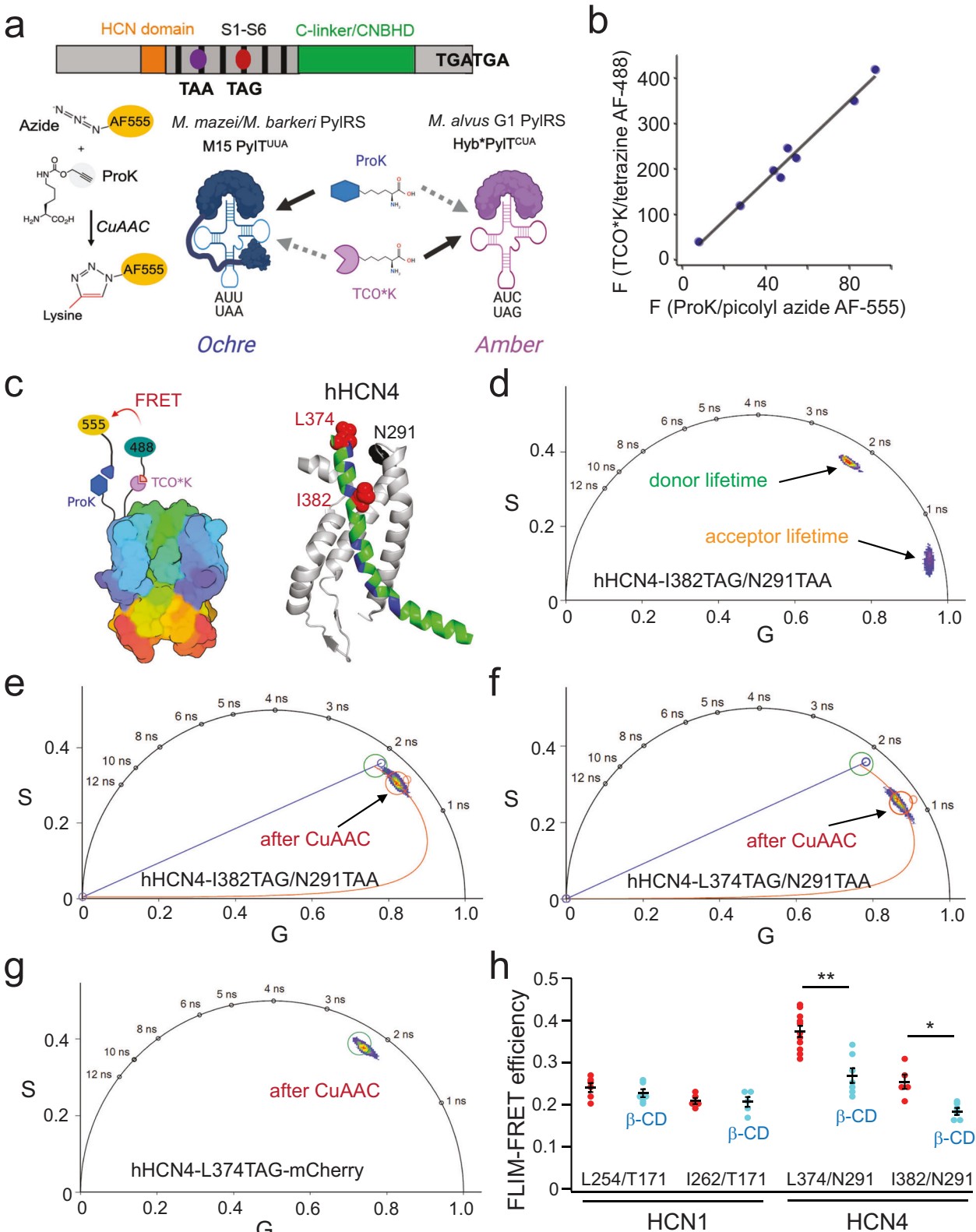

the FRET donor by $Cu^{2+}$–DOTA is ~5%, which aligns with the degree of nonspecific quenching observed with cyclen-bound transition metals, as previously reported (Supplementary Fig. 6)[2,16]. Furthermore, the acute application of 5 mM β-CD to deplete cholesterol reduced the FRET efficiency of the hHCN1-L254TAG/T171TAA pair of human hHCN1 whereas the β-CD application increased the FRET of the FRET pairs of hHCN4 channels (Fig. 5e and g). Again, this suggests that disrupting the

$L_o$ domains produced distinct effects on the VSD of HCN channels. In the resting state, our analysis demonstrated a comparatively minor alteration in the relevant tmFRET change for HCN1 in contrast to HCN4. Specifically, the hHCN1-I262/T171 pair did not exhibit statistical significance, implying that the impact of β-CD remains notably more significant for HCN4 when compared to HCN1. This dependence on membrane lipid domains suggests that the isoform-specific

**Fig. 4 | Amber/ochre dual stop-codon suppression enables fluorescence labeling for FLIM-FRET in living cells. a** Strategies for incorporating and labeling two different amino acids orthogonally using the copper(I)-catalyzed azide-alkyne cycloaddition (CuAAC), along with the IEDDA. **b** Linear correlation of the labeled AF488 versus AF555 fluorescence using the hHCN4 L374TAG/N291TAA construct. **c** Cartoon showing the design of the FRET pairs at the VSD of HCN channels. FRET sites were chosen for the HCN4 channel. **d** Phasor-plot presentation for the measured lifetimes of both the FRET donor AF-488 labeled to the HCN4-I382 site and the FRET acceptor AF-555 labeled to the HCN4-N291TAA site. **e** and **f** FRET was detected after CuAAC for the construct containing the ochre site; FRET in the hHCN4-L374TAG/N291TAA pair was greater than that in the hHCN4-I382TAG/

N291TAA pair. **g** FRET was not detected after CuAAC for the construct without the ochre site, hHCN4-L374TAG-mCherry. **h** Effects of the 5 mM β-cyclodextrin on the measured FRET using the phasor FLIM approach in the specified FRET pairs in the VSD of hHCN1 and hHCN4 channels. Data shown are mean ± s.e.m. For the HCN1 L254/T171: $n = 6$ cells for both conditions; for the HCN1 I262/T171: $n = 5$ cells for both conditions; for the HCN4 L374/N291: $n = 11$ cells for the control and $n = 7$ cells after β-CD; for the HCN4 I382/N291: $n = 5$ cells for the control and $n = 6$ cells after β-CD. One-way ANOVA, followed by Tukey's post hoc tests and pairwise comparisons (no adjustment), was used as the statistical analysis method; the highlighted statistical significance is $p = 7.1E{-}05$ for HCN4 L374/N291 and $p = 0.035$ for HCN4 I382/N291; $*p < 0.05$, $**p < 0.01$.

localization of HCN channels could directly impinge on the rearrangement of the S4 helix voltage sensor for its activation and the resulting channel opening.

We observed that the disruption of lipid domains had a significant impact on the change in FRET. However, when the membrane was depolarized using 120 mM KCl (Fig. 5d–h), the aforementioned effect was largely diminished. Using the same FRET pairs, we compared the FRET levels between imaging the membrane under 120 mM KCl and control conditions. For the hHCN1-L254TAG/T171TAA and hHCN4-L374TAG/N291TAA pairs, the FRET level decreased, whereas for the hHCN1-I262TAG/T171TAA and hHCN4-I382TAG/N291TAA pairs, the FRET level increased (Fig. 5d and h). This difference is likely attributed to the upward movement of the voltage sensor following depolarization, causing a shift in the FRET donor position relative to the acceptor. Furthermore, by employing the voltage-sensitive dye fVF2[59,60], we quantified that the application of 120 mM KCl resulted in a $68 \pm 8$ mV increase in voltage (Supplementary Fig. 4 and see the "Methods" section). This voltage change was sufficient to depolarize the membrane of tsA cells to above 0 mV. Notably, the effect caused by β-CD is more pronounced at more hyperpolarized voltages, corresponding to the activation state of HCN channels. Moreover, the tmFRET efficiencies were converted to distances using the Förster convolved Gaussian (FCG) relation as previously described (see the "Methods" section, Fig. 5b, and Supplementary Table 1)[2,56]. Taken together, these findings suggest that the voltage sensor of HCN channels is directly regulated by the isoform-specific localization to membrane domains, and this regulation is voltage-dependent.

## Discussion

Voltage-gated ion channels (VGICs) in the nervous and cardiac systems facilitate rapid and precise transmission of electrical signals essential for various physiological processes. This study tested the direct dependence of the voltage-sensor conformations on compartmentalized membrane domains, as well as their susceptibility to modulation by pathophysiological conditions that disrupt ordered membrane domains. Using dual stop-codon suppression to incorporate noncanonical amino acids, orthogonal fluorescence labeling via click chemistry, and conformational change measurement of VSD sites using transition metal FRET, we propose a hypothesis regarding a hydrophobic mismatch between the S4 helix voltage sensor of pacemaker HCN channels and the lipid bilayer (Fig. 6). This mismatch could enhance the sensitivity of the voltage sensor to hyperpolarizing voltages when in disordered domains. Importantly, this mechanism is reliant on the isoform-specific localization of HCN channels in lipid domains, with hHCN4 preferentially residing in ordered membranes, experiencing a lesser degree of hydrophobic mismatch compared to hHCN1 channels. As indicated by several structural models of the HCN VSD within different contexts, including detergents[2,14,15], homogeneous model membranes, and native membranes, the tilting motion of the S4 helix is influenced by the lipid environment (Fig. 6 and Supplementary Fig. 8). Particularly, we propose that this helical tilt is more pronounced within $L_d$ domains compared to $L_o$ domains (Fig. 6 and

Supplementary Fig. 8). In addition, the measured difference in tmFRET produced by β-CD between hHCN1 and hHCN4 exhibits a dependence on specific functional state, indicating that only the activated state of the S4 helix voltage sensor, occurring at more hyperpolarized voltages, is susceptible to modulation through localization to membrane lipid domains. Furthermore, alterations in the membrane fluidity and curvature, line tension, and osmotic water permeability may coincide with changes in membrane thickness and lipid orderliness[27,61–63]. These factors could potentially modulate the hydrophobic mismatch between lipid domains and HCN channels. To validate these concepts, further investigation employing techniques such as high-resolution electron microscopy and patch-clamp fluorometry is needed.

The mechanism underlying the isoform-specific lipid-domain localization of HCN channels remains unclear. Membrane proteins often use reversible palmitoylation of cysteines to localize themselves to lipid domains. HCN1, 2, and 4 channels possess multiple cysteines in their intracellular amino- and carboxyl-terminal regions[64]. Prior studies demonstrated that mutations of the amino-terminal palmitoylated cysteines have minimal effects on the voltage sensitivity of HCN2 and HCN4 channels[64]. These mutations may not explain the shift in the conductance-voltage relationship observed when disrupting the $L_o$ domains using β-CD. Additionally, all HCN channels contain a putative caveolin binding domain (CBD) located within the amino-terminal HCN domain and S1 helix[65]. It has been proposed that the binding of hHCN4 channels to caveolins serves as a mechanism for the localization of hHCN4 channels to membrane signaling domains, particularly caveolae in cardiac pacemaker cells[40,65]. However, the specific structural details of the potential interaction between HCN and caveolin (directly or indirectly through other scaffold proteins) have not been thoroughly characterized. Since the putative CBD is also present in hHCN1 channels, alternative mechanisms may account for the isoform-specific lipid domain localization of HCN channels. Notably, the amino- and carboxyl-terminal regions of hHCN4 channels differ considerably from those of hHCN1 channels, which could enable distinctive interactions with membrane proteins that are unique to specific isoforms.

The dual-codon suppression strategy, employing the orthogonal tRNA/tRNA synthetases from *M. alvus* and *M. mazei/M. barkeri* archaea have significantly improved the feasibility of conducting FRET experiments in living cells. This approach circumvents the need for cysteine-mediated labeling and mitigates the issue of nonspecific quenching of the FRET donor. Several future advancements can further refine this strategy. First, the development of PylRS-based fluorescent ncAAs that bring the fluorophore closer to the protein backbone is highly desirable, to facilitate measuring absolute distances. Second, efforts should be made to minimize the unintended suppression of stop codons, especially ochre codons, in endogenous proteins, as this would improve cellular health. Lastly, the creation of a faster and copper-independent azide-alkyne click chemistry, which can operate independently of the IEDDA reaction, would facilitate fluorescence labeling. Leveraging a wide range of

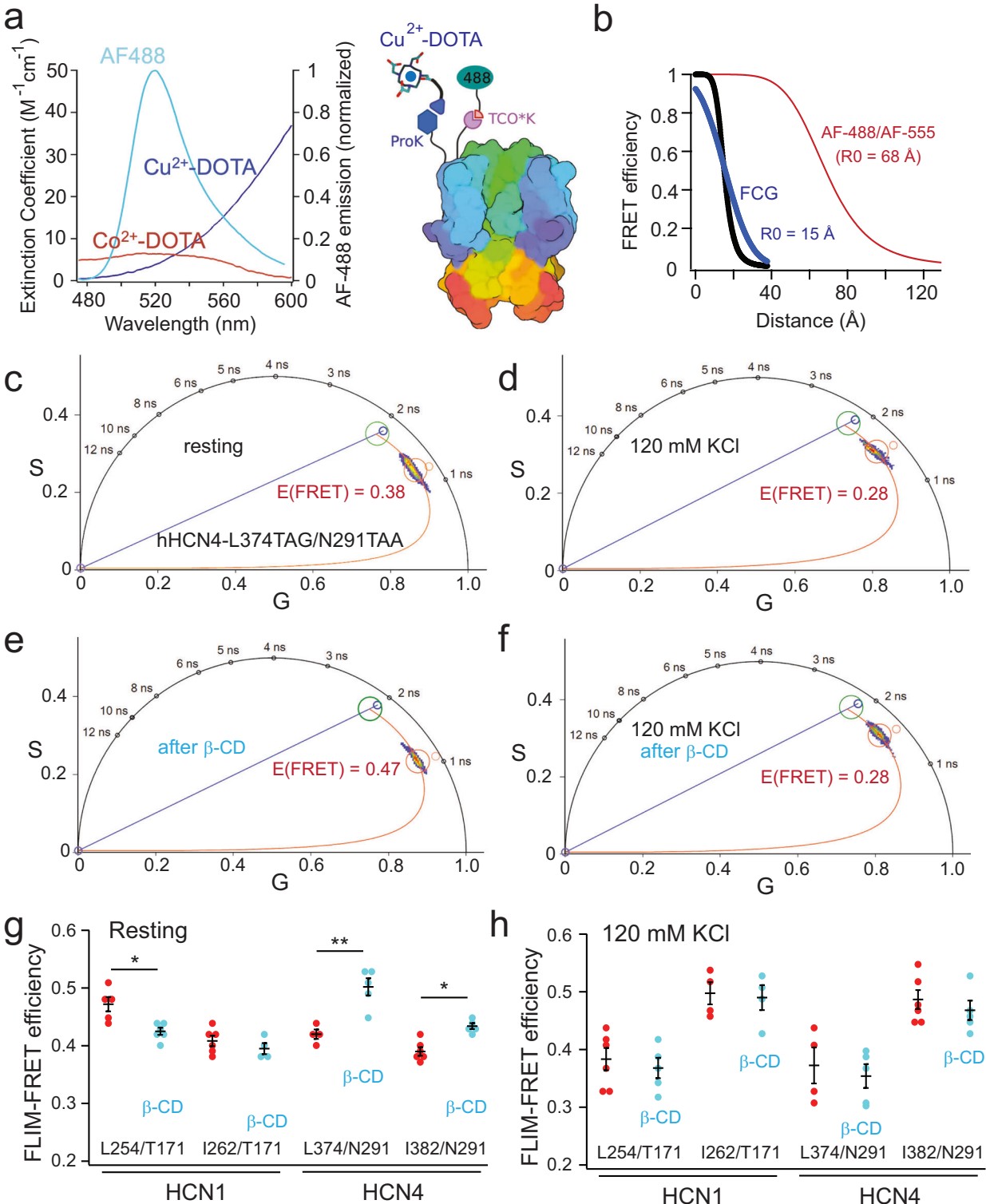

bright and photostable dyes available for click chemistry, the dual-codon suppression approach will propel the progress of structural studies in ion channel proteins using single-molecule and super-resolution fluorescence techniques.

Membrane lipids play a crucial role in controlling the function of ion channels, acting as key regulators[27,66,67]. However, the significance of compartmentalizing ion channels within lipid domains to regulate their voltage sensing has been largely overlooked, despite its implications for various disease states. Notably, neurodegeneration and inflammation have been observed to alter the composition and organization of membrane lipid domains[29,32,35,37,43,68]. Thus, the rearrangement of the S4 helix voltage sensor of HCN channels within specific membrane domains emerges as a potential contributor to neurological and cardiac disorders characterized by abnormal rhythmic firing of action potentials[6,69]. Conditions such as neuropathic pain, epileptic seizures, and cardiac arrhythmia are likely influenced by this membrane domain-dependent rearrangement. We recognize that changes in membrane lipid domains can impact the proper function of pacemaker channels and contribute to disease progression.

**Fig. 5 | Applying the transition metal FRET to measure the structural rearrangement of the VSD of HCN channels caused by lipid-domain disruption.**
**a** Spectral overlap between the AF-488 emission and the absorption spectra of $Cu^{2+}$ or $Co^{2+}$ chelated by azido-DOTA and the cartoon display of the tmFRET design.
**b** Distance dependence of FRET efficiency based on the Förster equation and the FCG relation ($\sigma = 10$ Å), highlighting a shorter $R_0$ of tmFRET, compared with that of the conventional FRET. **c–f** Phasor plot showing the shortening of the donor lifetime after the $Cu^{2+}$–DOTA reaction with the incorporated ProK for the hHCN4-L374TAG/N291TAA pair and conditions with 5 mM β-cyclodextrin and/or 120 mM KCl applications as indicated. **g** Effects of the 5 mM β-cyclodextrin on the measured tmFRET using the phasor FLIM approach in the specified FRET pairs in the VSD of hHCN1 and hHCN4 channels. Data shown are mean ± s.e.m. For the HCN1 L254/T171: $n = 5$ cells for the control and $n = 6$ cells after β-CD; for the HCN1 I262/T171: $n = 6$

cells for the control and $n = 4$ cells after β-CD; for the HCN4 L374/N291: $n = 4$ cells for the control and $n = 5$ cells after β-CD; for the HCN4 I382/N291: $n = 6$ cells for the control and $n = 5$ cells after β-CD. **h** Effects of the 5 mM β-cyclodextrin, in the presence of 120 mM KCl, on the measured tmFRET of the specified FRET pairs as in panel **g**. Data shown are mean ± s.e.m. For the HCN1 L254/T171: $n = 6$ cells for the control and $n = 5$ cells after β-CD; for the HCN1 I262/T171: $n = 6$ cells; for the HCN4 L374/N291: $n = 4$ cells for the control and $n = 5$ cells after β-CD; for the HCN4 I382/N291: $n = 6$ cells for the control and $n = 5$ cells after β-CD. One-way ANOVA, followed by Tukey's post hoc tests and pairwise comparisons (no adjustment), was used as the statistical analysis method; the highlighted statistical significance is $p = 0.01$ for HCN1 L254/T171, $p = 1.8\text{E}{-}04$ for HCN4 L374/N291, and $p = 0.018$ for HCN4 I382/N291 in the panel **g**; *$p < 0.05$, **$p < 0.01$.

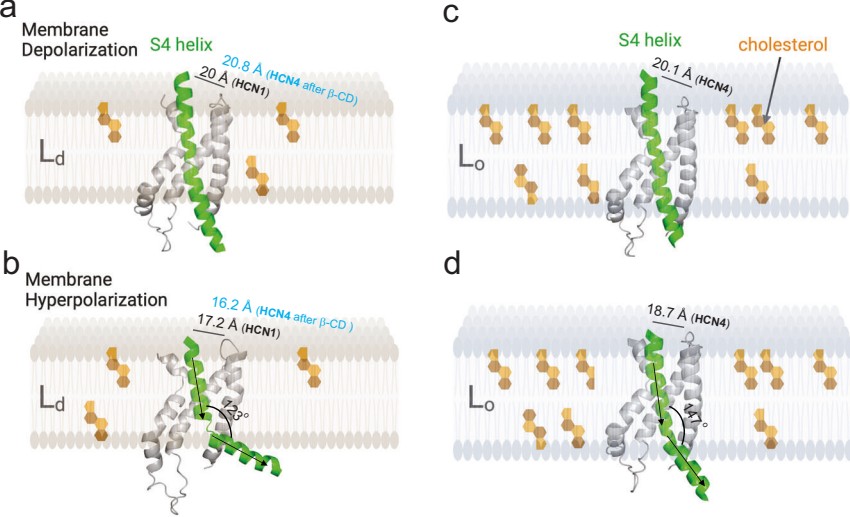

**Fig. 6 | Proposed structural model for the lipid domain-dependent conformational rearrangements of HCN voltage-sensing domains. a** and **b** Structural cartoons illustrate the rearrangement of the voltage-sensing domain (VSD), specifically the S4 helix (depicted in green), within HCN channels localized in thinner disordered phospholipid-dominant Ld domains. These structural changes are based on a molecular dynamics simulation model of hHCN1 conducted in a lipid bilayer composed solely of 1-palmitoyl-2-oleoyl-sn-glycero-3-phosphocholine (POPC)[15]. In comparison to these conformations, we modeled the hHCN4 VSD (SWISS modeling) based on a previous HCN-VSD model created using tmFRET

measurements in native lipid membranes combined with RosettaCM modeling[2], and hypothesize that it mimics the VSD within thicker ordered sphingolipid- and cholesterol-rich $L_o$ domains (depicted in panels **c** and **d**). We propose that the S4 helix exhibits a different conformation that facilitates its downward and tilting motion, within $L_d$ domains, thereby leading to channel hyperactivity (also see Supplementary Fig. 8). The distances illustrated are tmFRET measured distances between the L254/T171 pair of hHCN1 or the L374/N291 pair of hHCN4. Notably, these lipid-domain-specific conformational variances predominantly manifest under hyperpolarized membrane voltages.

## Methods

### Molecular cloning

The incorporation of non-canonical amino acids (ncAAs) for bioorthogonal labeling was done site-specifically through dual or single codon suppression using amber (TAG) and ochre (TAA) codons on the channel protein of interest: hHCN1 and hHCN4 cDNAs synthesized by Vector Builder (Chicago, IL). Constructs were cloned and amplified from a mammalian gene expression vector containing a carboxyl-terminal YFP or mCherry and a CMV promoter. DNA oligo primers were designed and synthesized by Integrated DNA Technologies (IDT, Coralville, IA). Point mutations were generated using the QuikChange II XL Site-Directed Mutagenesis or Multi Site-Directed Mutagenesis Kit (Agilent Technologies, Cedar Creek, TX) according to the manufacturer's instructions. The oligo primer sequences are provided in the Supplementary Table 2. The DNA plasmids were treated with Dpn I restriction enzyme (Agilent Technologies) to digest the methylated parental DNA template and subsequently transformed into XL10-Gold competent cells (Agilent Technologies). The functioning stop codon, opal (TGA), was introduced with the deletion of YFP. The presence of the dual stop codons and functioning stop codons in the mutant plasmid was verified through DNA sequencing provided by Genewiz (Azenta Life Sciences, Burlington, MA) or Plasmidsaurus (Eugene, OR).

In addition, DNA concentration was measured using a Nanodrop OneC spectrophotometer (Life Technologies, Grand Island, NY).

### Cell culture, transient transfection, and bioorthogonal labeling

Frozen aliquots of tsA-201 cells (a variant of human embryonic kidney HEK cells) were obtained from Sigma-Aldrich (St. Louis, MO), authenticated using STR profiling, and stored in liquid nitrogen. At the start of a new cell line, cells were cultured in Dulbecco's modified Eagle's medium (DMEM; Gibco) supplemented with 10% fetal bovine serum (FBS; Gibco) and 1% penicillin/streptomycin (Gibco) in a humidified incubator at 37 °C with 5% $CO_2$ in tissue culture dishes (CellTreat, Pepperell, MA).

Transfection was performed on 70–90% confluent tsA-201 cells using the lipofectamine 3000 kit (Invitrogen, Carlsbad, CA, #L30000080), following previously established protocols[43,70]. For dual stop-codon suppression, the cells were co-transfected with two plasmids: pAS_4xU6-PyIT M15(UUA) FLAG-Mma PyIRS (Mma PyIT/RS) and pAS_4xhybPyIT A41AA C55A FLAG-G1 PyIRS Y125A (G1PyIT/RS) tRNA/aminoacyl-tRNA synthetase pairs from Addgene (Watertown, MA, #154774 and #154773, respectively), along with a third plasmid containing the dual stop-codon mutations required for selective incorporation of two noncanonical amino acids (ncAAs): trans-

cyclooct-2-ene-ʟ-lysine (TCO*K) and N-propargyl-ʟ-lysine (ProK) (SiChem, Bremen, Germany), as described previously[49]. For single stop-codon suppression, the pAS_4xG1 PylT FLAG-G1 PylRS Y125A tRNA/aminoacyl-tRNA synthetase plasmid (Addgene, Watertown, MA #154773) was used. The ncAAs were added to the transfection solution: 0.1 mM TCO*K and 0.25 mM ProK, and mixed with 1 M HEPES solution in a 1:4 ratio (Sigma-Aldrich, St. Louis, MO) to cell culture. Stock solutions (100 mM) were made by dissolving TCO*K and ProK in 0.2 M NaOH water.

Two-color bioorthogonal labeling was employed using click chemistry, following an established method[49]. At the TCO*K site, strain-promoted inverse electron-demand Diels–Alder cycloaddition (SPIEDAC) was performed by adding 1–2 μM tetrazine dye (Click Chemistry Tools, Scottsdale, AZ) to the culture medium and incubating it with the cells at 37 °C for 30 min. At the ProK site, the copper-catalyzed azide-alkyne cycloaddition (CuAAC) reaction was carried out using 50 μM $CuSO_4$, 250 μM THPTA (Tris(benzyltriazolylmethyl) amine), and 2.5 mM ascorbic acid. The specific labeling procedure varied depending on the azide dye used to react with the ProK. Cells selectively labeled with 5 μM picolyl azide dye (Click Chemistry Tools, Scottsdale, AZ) were incubated for 10 min at room temperature. For cells labeled with azido DOTA (50 μM in cell culture), the incubation was performed for 5 h at 37 °C. To prepare $Cu^{2+}$–DOTA, 10 μL each of 100 mM DOTA azide stock and 110 mM $CuSO_4$ stock were mixed and allowed to incubate for 5 min as the solution turned from light to deeper blue, indicating binding of $Cu^{2+}$ to the DOTA. Once bound, DOTA and $Cu^{2+}$ dissociation is very slow due to their picomolar affinity. The $Cu^{2+}$-DOTA complex is stable and is not in conflict with the CuAAC. We found that the mild reducing agent ascorbic acid did not affect the absorption of $Cu^{2+}$-DOTA at the visible light range (see Supplementary Fig. 6). The azido-mono-amide-DOTA was purchased from Macrocyclics, Inc (Plano, TX).

Furthermore, the incorporation of L-Anap was conducted as previously described[56], using 20 μM L-Anap methyl ester (AsisChem, Waltham, MA) in the cell culture. To enhance nonsense suppression, a human dominant-negative release factor eRF1 (E55D) (Vector Builder, Chicago, IL) was co-transfected[47]. Before imaging, cells were cultured in the ncAA-free DPBS buffer for at least 10 min, to gently washing away unincorporated ncAA. Regarding the membrane permeable methyl-ester form of L-Anap that was employed for its incorporation, note that L-Anap might have exhibited non-specific interactions with the membrane even after the washing step. This could potentially contribute to an underestimation of the observed change in the lifetime of the incorporated L-Anap.

The Cholera Toxin Subunit B-Alexa Fluor™ 594 conjugate was obtained from Thermo Fisher Scientific and administered to cultured cells at a concentration of 20 nM for a duration of 10–15 min. The binding affinity of CTxB to GM1-enriched membrane falls within the picomolar range[26].

## Fluorescence microscopy, spectral FRET, and spectrophotometry

Spectral Förster resonance energy transfer (FRET) experiments were conducted following a methodology similar to a previous paper[43]. Fluorescence measurements were performed using an Olympus IX73 inverted microscope (Olympus America, Waltham, MA) equipped with a ×60 U Plan S-Apo 1.2-NA water-immersion objective. Epifluorescence recording was achieved using wide-field excitation from a pE-800 LED light source (CoolLED, Andover, UK). For spectral measurements, an imaging apparatus was employed, which consisted of a spectrograph (SpectroPro 2156, 300 groves/mm grating, 500 nm blaze; Teledyne Princeton Instruments, Acton, MA) placed between the microscope output port and an INFINITY3S-1URM CCD camera (Teledyne Lumenera, Ottawa, ON). CFP fluorescence was excited using a 435 nm LED line along with a 436/20 nm excitation filter within the filter cube.

Similarly, YFP and AF-488 were excited using a 500 nm LED line along with a 500/20 nm excitation filter within their respective filter cube. Emission filters were removed from the filter cubes, and the spectrograph functioned as an emission filter, allowing only fluorescence in the wavelength range of 423–597 nm, as per the manufacturing calibration (Teledyne Princeton Instruments). The long-pass dichroic mirrors with cutoff wavelengths of approximately 455 and 515 nm were used in the CFP and YFP/AF-488 filter cubes, respectively. Spectral images were acquired with exposure times of 200–500 ms, adjusted based on the fluorescence intensity of each individual cell, using the INFINITY3S-1URM CCD camera and the Infinity Analyze and Capture software (Teledyne Lumenera). The exposure time remained consistent while recording emission spectra for both donors and acceptors within the same single-cell FRET experiment.

Subsequent analysis of the spectral data was conducted using ImageJ/Fiji (National Institutes of Health) and Igor Pro 9 (Wavemetrics). To account for background fluorescence, spectra were obtained by subtracting signals from areas outside the cell of interest. The solution used for cell imaging was Dulbecco's phosphate-buffered saline (DPBS, ThermoFisher) unless otherwise specified. All microscopic experiments were carried out at room temperature.

FRET experiments using donor-only constructs (CFP) were performed to acquire the averaged emission spectrum of the donor when excited at the donor excitation wavelength. This emission spectrum of the donor alone, termed $F_{CFP}^{CFP}$, was subsequently scaled and normalized to the peak intensity of the $F_{CFP/YFP}^{CFP}$ spectrum for each experiment. In the labeling convention here, superscript denotes the specific property of the excitation light, either pertaining to the donor or acceptor. Subscript, on the other hand, indicates the fluorophores expressed in the experiment, which could be both donor and acceptor, donor alone, or acceptor alone. To generate the spectrum FFRET, the normalized $F_{CFP}^{CFP}$ spectrum was subtracted from the $F_{CFP/YFP}^{CFP}$ spectrum. This subtraction was carried out using the following equations:

$$F_{FRET} = F_{CFP/YFP}^{CFP} - F_{CFP}^{CFP} \qquad (1)$$

This subtraction was for correcting the bleed-through of the donor emission into the spectral range for the acceptor emission. Ratio A was then calculated as

$$\text{Ratio } A = F_{FRET} / F_{CFP/YFP}^{YFP} \qquad (2)$$

The denominator of the Ratio A is the spectrum generated by using the excitation light at the YFP excitation wavelength. In addition, two emission spectra were collected from cells that were only transfected with YFP-containing constructs: $F_{YFP}^{CFP}$, which was the emission spectrum using the donor (CFP) excitation; and $F_{YFP}^{YFP}$, which was the emission spectrum using the acceptor (YFP) excitation. Ratio A0 was calculated using:

$$\text{Ratio } A0 = F_{YFP}^{CFP} / F_{YFP}^{YFP} \qquad (3)$$

Ratio A0 measures the efficiency of the crosstalk of FRET, which contains the fraction of the direct excitation of FRET acceptors by the donor-excitation light. Ratio A0 is a constant of 0.124 for the CFP/YFP pair under our experimental conditions. The FRET efficiency was linearly proportional to the difference between Ratio A and Ratio A0[71]. The apparent FRET efficiency ($E_{app.}$) was calculated using the following equation:

$$E_{app.} = \frac{\text{Ratio } A - \text{Ratio } A0}{\text{Ratio } A0} * \frac{\varepsilon A}{\varepsilon D} \qquad (4)$$

The molar extinction coefficients of the acceptors ($\varepsilon$A) and donors ($\varepsilon$D) at the donor excitation wavelength were used in the calculations. The specific values for the molar extinction coefficients (or molar absorptivity) were obtained from FPbase.org. The extinction coefficient for YFP was found to be 83,400 $\text{cm}^{-1}\text{M}^{-1}$, while the value for CFP was determined to be 32,500 $\text{cm}^{-1}\text{M}^{-1}$. Considering the known absorption spectrum of YFP, the ratio of YFP absorption at the CFP excitation wavelength relative to its peak absorption was found to be 0.04. For the CFP/YFP pair, the ratio of $\varepsilon$A to $\varepsilon$D was estimated to be 0.10.

We found a positive correlation between the apparent FRET efficiency ($E_{app.}$) and the ratio of peak fluorescence intensity between the FRET donor and acceptor, denoted as $F = F_{donor}/F_{acceptor}$. When the $F$ value is very high, it is anticipated that the $E_{app.}$ will reach a saturation point and stabilize at a steady-state value, which corresponds to the maximum FRET efficiency $E_{max.}$[71]. As previously described[43], the following equation was used for fitting the relationship between the apparent FRET efficiency and the $F$:

$$E_{app.} = E_{max.} * \frac{F}{S+F} \qquad (5)$$

In this equation, the coefficient $S$ is dependent on multiple factors, including the optical properties of the fluorophores, the recording system, the proportion of fluorophores involved in close-range FRET interactions, and the efficiency of FRET itself. We performed fittings to determine the parameters, and the chi-square goodness of fit test was used for evaluation. Importantly, all the obtained chi-square values were below 0.1, indicating a good fit between the experimental data and the equation.

The absorption of transition metals was measured using the Nanodrop OneC spectrophotometer (Life Technologies, Grand Island, NY), a full spectrum micro-volume UV–Vis spectrophotometer with a 190–688 nm wavelength measurement range.

**Fluorescence lifetime measurement and FLIM-FRET**

Digital frequency-domain fluorescence lifetime imaging[72] was carried out using a Q2 laser scanning confocal system equipped with a FastFLIM data acquisition module (ISS, Inc., Champaign, IL), and two hybrid PMT detectors (Model R10467-40, Hamamatsu, USA, Bridgewater, NJ). The confocal FLIM system is coupled to an Olympus IX73 inverted microscope (Olympus America, Waltham, MA). The fluorescence lifetime measurements include both the frequency-domain (instantaneous distortion-free phasor plotting) and time-domain decay information. To excite L-Anap, a 375 nm pulsed diode laser (ISS, Inc.) driven by FastFLIM was employed. The emitted fluorescence from L-Anap was captured using a 447/60 nm band-pass emission filter. Simultaneous two-color imaging was facilitated by using dichroic cubes, one containing a 50/50 beam splitter and the other equipped with various long-pass and band-pass filters. For all other fluorophores, except for L-Anap, a supercontinuum laser (YSL Photonics, Wuhan, China) with a wavelength range of 410–900 nm, a repetition rate of 20 MHz, and a pulse duration of 6 ps was used for fluorescence excitation. Specifically, CFP was excited at 442 nm wavelength, AF-488 and YFP at 488 nm wavelength, and AF-555, AF-594, or mCherry at 532 or 561 nm wavelength. To detect the emission from the CFP/YFP FRET pair, an emission filter cube comprising a 475/28 nm filter for CFP, a 495 nm long-pass dichroic mirror, and a 542/27 nm filter for YFP was employed. For the AF-488/AF-555 or AF-488/AF-594 pairs, an emission filter cube consisting of a 525/40 nm filter for AF-488, a 552 nm long-pass dichroic mirror, and a 593/40 nm filter for AF-555, AF-594. The mCherry emission was measured also using the 593/40 nm filter.

For image acquisition, confocal images of 256 × 256 pixel frame size were obtained under each experimental condition, with a motorized variable pinhole (100 μm used) and a pixel dwell time of

0.1 ms. The acquired pixels were subsequently examined using the phasor plot. The VistaVision software, equipped with the imaging and FLIM module, facilitated image processing, display, and acquisition, allowing for the specification of parameters such as pixel dwell time, image size, and resolution. Furthermore, the fitting algorithm and multi-phasor analysis were employed for FLIM analysis. A median/gaussian smoothing filter and an intensity-threshold filter were applied to improve the display and the analysis of membrane-localized lifetime species. Phasor FLIM has the advantage of distinctly differentiating between signals originating from the background outside of the membrane and those within the plasma membrane. Consequently, the process of excluding background interference outside of the membrane becomes significantly more straightforward compared to conducting measurements reliant on fluorescence intensity.

To determine the phase delays ($\varphi$) and modulation ratios ($m$) of the fluorescence signal in response to an oscillatory stimulus with frequency $\omega$, the VistaVision software was used. These values were obtained by performing sine and cosine Fourier transforms of the phase histogram, taking into account the instrument response function (IRF) calibrated with Atto 425 in water (with a lifetime of 3.6 ns), rhodamine 110 in water (with a lifetime of 4 ns) and rhodamine B in water (with a lifetime of 1.68 ns), depending on the emission spectrum of the fluorophore of interest.

FLIM-FRET analysis was conducted using the FRET trajectory function available in the VistaVision software[44]. To obtain accurate results, three key parameters were adjusted to optimize the fitting trajectory through both the unquenched donor and the donor species caused by FRET. These parameters were as follows: (1) background contribution in the donor sample, (2) unquenched donor contribution in the FRET sample, and (3) background contribution in the FRET sample. The background levels in the donor sample and the FRET samples were set below 4%. This value aligns with the very low emission intensity observed in untransfected cells compared to those with overexpressed proteins. Alternatively, the FLIM-FRET efficiency was calculated using the following equation after performing a double-exponential fitting of the time domain FLIM data:

$$E_{FRET} = 1 - \frac{\tau(DA)}{\tau(D)} \qquad (6)$$

where the $\tau(DA)$ represents the amplitude-weighted average lifetime of the FRET donor in the presence of the acceptor, and the $\tau(D)$ represents the lifetime of the unquenched donor in the absence of acceptor. The FRET efficiency obtained from the FRET trajectory function of the phasor plot was compared with the FRET efficiency calculated by Eq. (6) with the lifetimes estimated from the exponential fitting analysis. This comparison yielded consistent results, indicating agreement between the two methods (Supplementary Fig. 7).

Furthermore, to correct the nonspecific quenching (around 5% by unlabeled $Cu^{2+}$-DOTA) for more accurate distance estimation, we used the following Eq. (9) by rearranging the Eqs. (6)–(8) to obtain a corrected $E_{FRET}$($E_{FRET, corrected}$).

$$E_{nonspecific} = 1 - \frac{\tau(D_{nonspecific})}{\tau(D)} \qquad (7)$$

$$E_{FRET,corrected} = 1 - \frac{\tau(DA)}{\tau(D_{nonspecific})} \qquad (8)$$

$$E_{FRET, corrected} = 1 - \frac{1 - E_{FRET}}{1 - E_{nonspecific}} \qquad (9)$$

where the $E_{nonspecific}$ is estimated using the construct that does not contain the FRET acceptor site by not including the ochre TAA

codon, e.g. HCN4-L374TAG-mCherry (Supplementary Fig. 6c, d) and using the FRET trajectory method of the phasor plot. The $\tau(D_{nonspecific})$ is the donor lifetime after nonspecific quenching in the absence of the transition metal-coordination site. These equations are derived mathematically by considering a kinetic scheme for a fluorophore relaxing from the excited state, according to previous research[48].

The distances $r$ between the tmFRET donors and acceptors were estimated using the convolution of the Förster equation $E_{max.} = 1/(1 + (r/R_0)^6)$ with a Gaussian function: Förster convolved Gaussian (FCG) as previously established[2,56]. The FCG relation was presumed to follow a Gaussian distribution with a standard deviation ($\sigma$) of 10 Å and a full width at half maximum (FWHM) of 23.548 Å. This assumption was derived from the interatomic distance distribution of probes with similar labels[56,73]. The $R_0$ value was calculated using the following equation:

$$R0 = C\sqrt[6]{JQ\eta^{-4}\kappa^2} \qquad (10)$$

where $C$ is the scaling factor, $J$ is the overlap integral of the Anap emission spectrum and the YFP absorption spectrum, $Q$ is the quantum yield of the AF-488 (0.92 according to FPbase), $\eta$ is the index of refraction, and $\kappa^2$ is the orientation factor. $\eta$ was assumed to be 1.33, and $\kappa^2$ was assumed to be 2/3.

### Patch-clamp electrophysiology

The whole-cell patch-clamp recordings were performed using an EPC10 patch-clamp amplifier and PATCHMASTER (HEKA) software. Borosilicate patch electrodes were created using a P1000 micropipette puller from Sutter Instrument (Sutter, Novato, CA), resulting in an initial pipette resistance of approximately 3–5 MΩ. Currents were sampled at a 10 kHz frequency. A Sutter MP-225A motorized micromanipulator was used for the patch clamp (Sutter, Novato, CA). The recordings were conducted at a temperature range of 22–24 °C.

To record HCN currents from tsA cells, the internal pipette solution was prepared with the following composition (in mM): 10 NaCl, 130 KCl, 3 MgCl$_2$, 10 HEPES, 0.5 EGTA, 3 MgATP, 0.3 Na$_3$GTP, and pH 7.2 adjusted with KOH. The external solution used was Ringer's solution with the following composition (in mM): 160 NaCl, 2.5 KCl, 1 MgCl$_2$, 2 CaCl$_2$, 10 HEPES, 8 D-Glucose, and pH 7.4 adjusted with NaOH. For measuring dual stop-codon suppression HCN channel currents, cell-attached patch-clamp recordings were employed instead of the whole-cell configuration due to the challenges associated with achieving a successful break-in (Supplementary Fig. 5).

The conductance–voltage (G–V) relationships were measured from the instantaneous tail currents at −40 mV following voltage pulses from 0 mV to between 0 and −140 mV. The leak tail currents following pulses to −40 mV were subtracted, and the currents were normalized to the maximum tail current following voltage pulses to −140 mV ($G/G_{max}$). The relative conductance was plotted as a function of the voltage of the main pulse and fitted with a Boltzmann equation:

$$G/G_{max} = 1/(1 + \exp[(V − V_{1/2})/V_s]) \qquad (11)$$

where $V$ is the membrane potential, $V_{1/2}$ is the potential for half-maximal activation, and $V_s$ is the slope factor.

### Statistics and reproducibility

The data parameters were presented as the mean ± s.e.m. of $n$ independent cells. Statistical significance was determined using the two-sided Student's $t$-test for comparing two groups, or a one-way ANOVA followed by Tukey's post hoc test for pairwise comparisons with more than two groups, denoted as $*p < 0.05$, $**p < 0.01$, respectively.

### Reporting summary

Further information on research design is available in the Nature Portfolio Reporting Summary linked to this article.

### Data availability

Please refer to https://doi.org/10.2210/pdb6UQF/pdb for the PDB accession code 6UQF used in this paper. Source data are provided with this paper.

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

## Acknowledgements
We thank Samantha Deavila for technical support, Bertil Hille for sharing L10 and S15 DNA plasmids, E. James Petersson for sharing fluorescent dyes, Joel C. Eissenberg and Enrico Di Cera for advice. Cartoons in figures were created using BioRender.

## Author contributions
G.D. conceptualized and designed the research. G.D. and L.J.H. performed experiments and analyzed data. G.D. wrote the manuscript.

## Competing interests
The authors declare no competing interests.
