## [Peer Review File · Nature Communications]

DIRECT REGULATION OF THE VOLTAGE SENSOR OF HCN CHANNELS BY MEMBRANE LIPID COMPARTMENTALIZATIONReviewers' Comments:

Reviewer #1:

Remarks to the Author:

In the manuscript "DIRECT REGULATION OF THE VOLTAGE-SENSING DOMAIN OF HCN CHANNELS BY MEMBRANE LIPID COMPARTMENTALIZATION" Handlin and Dai address the physiologically important and scientifically interesting question about how specific lipid domains within cellular membranes affect membrane protein function. The authors use two members of the HCN channel family, which is clever, as both members show different localization patterns and thus allow for direct comparison of the results obtained in ordered or disordered membrane compartments and to draw conclusions about differential regulation in cells.

By employing a variety of FRET techniques, they produce sound data with the necessary controls. The combination of FLIM-FRET and dual incorporation of non-canonical amino acids is novel and will be of interest to a broad audience in the field of membrane protein biophysics. Furthermore, the manuscript is well written and a pleasure to read, despite its complexity.

I have no concerns regarding the experimental approach. However, I would like to raise a few points related to the interpretation of the results and the presentation of the proposed mechanism.

The title states "Direct regulation of the VSD...". This interpretation is based on the depletion of cholesterol from the membrane using cyclodextrin which breaks up lipid domains, which the authors correlate with membrane thickness. They do not take into account effects on fluidity, curvature, water penetration and compressibility. Especially the last two effects, however, might be relevant. Water penetration into the membrane has the potential to ablate voltage sensing of positively charged residues on S4. Compressibility of the membrane, on the other hand, should also be important when talking about hydrophobic mismatch of the membrane and the protein. The authors should consider expanding their interpretation and discussion to such effects.

Secondly, the authors only test for S4 movements. Thus, in my opinion, talking about the VSD (S1-S4) is an over-interpretation, which can be easily fixed by more precise wording. A more complex question is whether other parts of the protein are similarly affected. The authors only look at the S4 helix. Accordingly, the interpretation naturally is related to this helix, but they do not test for other parts of the trans-membrane domain. Lipids might not directly regulate the VSDs but might affect the protein in general (including, for example, the pore domain, or by altering overall protein dynamics). The best solution would be to perform similar experiments with proteins labelled at different positions covering the entire TM domain (S1-S6). I understand that this comes with a considerable workload, which I would appreciate to see but will not request. However, in the absence of these additional data, the interpretation throughout the manuscript should be limited to the S4 helix. In this case, more emphasis could be put on the technological/methodological advances.

Lastly, Figure 6 could be improved, and the description of the proposed mechanism could be more extensive. Measured/ estimated distances, directionality of the movements, and angles etc could all be included in the figure. Since differences in FLIM-FRET are only observed under hyperpolarized conditions, state-specific effects should be discussed, meaning the resting state of HCN channels is not affected by localization to membrane compartments but rather the open state or the closed-open transition.

Reviewer #2:

Remarks to the Author:

The manuscript by Handlin and Dai uses extensive and highly quantitative FRET measurements to probe the effects of HCN channel localization in distinct membrane domains. The authors report that

HCN4 channels preferentially localize to ordered domains compared to HCN1 channels. Disruption of membrane domains affects FRET measurements suggesting voltage-sensing domain movement may depend on the lipid domains.

The study is impressive in technical accomplishments (dual-codon suppression, confocal FRET-FLIM), however, there are a few key limitations as noted below. Briefly, although the authors show that disruption of lipid domains affects voltage-sensor movement (a finding that is not surprising as previous studies have shown that beta-CD affects voltage-dependence of activation), the exact nature of this conformational change is unclear. There is also no clear mechanistic insight on factors that contribute to differential localization of HCN1 versus HCN4 in distinct lipid compartments.

Major concerns:

1. The authors assert that "HCN4 channels exhibit a greater affinity for ordered domains compared to HCN1 channels." I am not sure I understand this. The spectral FRET data shows that there is a similar FRET relationship between HCN4 and L10 probe (marking Lo) and S15 probe (Ld), while HCN1 has a high FRET relationship with S15 but not L10. To me, this seems to suggest that HCN4 is somewhat non-specific in preferring ordered and disordered domains, while HCN1 is preferentially excluded from ordered domains. Also it is worth noting that there is no real measurement of affinity here.

Furthermore, how does one exclude the possibility that HCN1 may have a different conformation in L10 domains that led to different amounts of FRET?

2. For a substantial number of figures, conclusions are drawn from quite small sample sizes, which I assume here refer to cells(?). In some cases, it is only $n = 3$. Is there sufficient power to draw statistical inferences with such small sample sizes?

3. In Fig. 3c, the authors show that a change in voltage also changes L-anap lifetime. However, if ordered domains are disrupted, this change vanishes. These data are then used to suggest that voltage sensor rearrangement in HCN4 may differ when membrane lipid domain is disrupted. The dataset is quite weak in my opinion. As noted in point 3, the sample size is quite limited. Furthermore, the axis limits are adjusted to artificially accentuate the changes. In the end though, L-anap elicits a very mild change in lifetime (<0.1 ns or $\sim 3\%$ change in signal). If you look closely, there is a reduction in lifetime of L-anap even in the presence of beta-cyclodextran. It is less prominent. Also, it may be helpful if the authors also performed similar experiments with HCN1. Presumably since it does not localize to ordered domains, the change in lifetime should be minimal.

4. Fig. 5g axis limits should be adjusted such that the minimum is 0 or at least a common value between resting and 120 KCl. There are other figures with the same issue.

5. "These findings imply that the conformation of the voltage-sensing domain (VSD) in HCN4 channels is influenced by membrane lipid domains. In contrast, HCN1 channels, predominantly found in disordered domains, appear to be less susceptible to changes induced by β -CD treatment" If this is the case, then, why is it that, disrupting the ordered domain in Fig. 5g show a strong reduction in FLIM-FRET efficiency similar to the change observed with HCN4 (albeit in reverse). One would've expected there to be no effect of beta-cyclodextran here.

6. Also in Fig. 4h, with HCN4, the FLIM-FRET efficiency decreases upon beta-CD. However, the change seems to opposite with tmFRET. Is there an explanation for this? I understand that tmFRET is more sensitive to within-subunit changes. Does this mean the intrasubunit probes get closer but the intersubunit FRET gets further? If this is the case, could you make a plausible case for what the intra versus inter subunit conformational change is?

7. Is membrane expression of HCN1 and HCN4 strongly reduced upon incorporating ncAAs following dual codon-suppression? Based on exemplar traces in Supplementary Fig. 1 versus 5, the amplitudes are 50-fold lower? Is this because the chemical modification affects channel trafficking or just reconstitution is low efficacy. Also, how does this affect FLIM-FRET measurement? For instance, voltage depolarization or hyperpolarization may only affect a subset of channels that are at the plasma

membrane and there could be substantial background signal from near membrane channels in vesicles?

8. Does beta-CD affect HCN1 versus HCN4 currents? If there is a change in VSD movement, one would envision there should be an appreciable change in channel biophysical properties for HCN4 channels but HCN1 should presumably be resistant to these? It may be helpful to determine the magnitude of changes with beta-CD in the specific expression system they are using since the amount and nature of lipid domains could vary depending on the expression system.

REVIEWER COMMENTS

Reviewer #1 (Remarks to the Author):

In the manuscript “DIRECT REGULATION OF THE VOLTAGE-SENSING DOMAIN OF HCN CHANNELS BY MEMBRANE LIPID COMPARTMENTALIZATION” Handlin and Dai address the physiologically important and scientifically interesting question about how specific lipid domains within cellular membranes affect membrane protein function. The authors use two members of the HCN channel family, which is clever, as both members show different localization patterns and thus allow for direct comparison of the results obtained in ordered or disordered membrane compartments and to draw conclusions about differential regulation in cells.

By employing a variety of FRET techniques, they produce sound data with the necessary controls. The combination of FLIM-FRET and dual incorporation of non-canonical amino acids is novel and will be of interest to a broad audience in the field of membrane protein biophysics. Furthermore, the manuscript is well written and a pleasure to read, despite its complexity.

We thank the reviewer for the thoughtful review with positive feedback.

I have no concerns regarding the experimental approach. However, I would like to raise a few points related to the interpretation of the results and the presentation of the proposed mechanism.

The title states “Direct regulation of the VSD...”. This interpretation is based on the depletion of cholesterol from the membrane using cyclodextrin which breaks up lipid domains, which the authors correlate with membrane thickness. They do not take into account effects on fluidity, curvature, water penetration and compressibility. Especially the last two effects, however, might be relevant. Water penetration into the membrane has the potential to ablate voltage sensing of positively charged residues on S4. Compressibility of the membrane, on the other hand, should also be important when talking about hydrophobic mismatch of the membrane and the protein. The authors should consider expanding their interpretation and discussion to such effects.

Thank you for these points. We have included these factors in the discussion of the paper.

Secondly, the authors only test for S4 movements. Thus, in my opinion, talking about the VSD (S1-S4) is an over-interpretation, which can be easily fixed by more precise wording. A more complex question is whether other parts of the protein are similarly affected. The authors only look at the S4 helix.

Accordingly, the interpretation naturally is related to this helix, but they do not test for other parts of the trans-membrane domain. Lipids might not directly regulate the VSDs but might affect the protein in general (including, for example, the pore domain, or by altering overall protein dynamics). The best solution would be to perform similar experiments with proteins labelled at different positions covering the entire TM domain (S1-S6). I understand that this comes with a considerable workload, which I would appreciate to see but will not request. However, in the absence of these additional data, the interpretation throughout the manuscript should be limited to the S4 helix. In this case, more emphasis could be put on the technological/methodological advances.

We have made revisions including changing the title of the manuscript to reflect that we were focusing on the voltage sensor S4 helix. The measured β -CD-induced rearrangements are between the S4 helix and the S1-S2 helix loop within the VSD. We acknowledge the point of performing an analysis of all TM helices, but as the reviewer mentioned, it may be outside of the scope of this paper and for a timely publication of this research.

We emphasized the technological advances in the Discussion.

Lastly, Figure 6 could be improved, and the description of the proposed mechanism could be more extensive. Measured/ estimated distances, directionality of the movements, and angles etc could all be included in the figure. Since differences in FLIM-FRET are only observed under hyperpolarized conditions, state-specific effects should be discussed, meaning the resting state of HCN channels is not affected by localization to membrane compartments but rather the open state or the closed-open transition.

We have improved the Fig. 6 by adding more descriptions and illustrations as well as adding a new Supplementary Fig. 8 for highlighting the differences in the directionality of the movements, and angles. In addition, the state-specific effect has been added to the discussion.

Reviewer #2 (Remarks to the Author):

The manuscript by Handlin and Dai uses extensive and highly quantitative FRET measurements to probe the effects of HCN channel localization in distinct membrane domains. The authors report that HCN4 channels preferentially localize to ordered domains compared to HCN1 channels. Disruption of membrane domains affects FRET measurements suggesting voltage-sensing domain movement may depend on the lipid domains.

The study is impressive in technical accomplishments (dual-codon suppression, confocal FRET-FLIM), however, there are a few key limitations as noted below. Briefly, although the authors show that disruption of lipid domains affects voltage-sensor movement (a finding that is not surprising as previous studies have shown that β -CD affects voltage-dependence of activation), the exact nature of this conformational change is unclear. There is also no clear mechanistic insight on factors that contribute to differential localization of HCN1 versus HCN4 in distinct lipid compartments.

We appreciate the reviewer's compliment on the technical advancement of this paper as well as the careful evaluation and insightful inquires of our manuscript.

We agree with the reviewer that there are still biophysical questions remain unanswered particularly about how exactly the voltage sensor rearranges differently in distinct lipid domains, its dynamics, and energetics. Currently, there are significant technical barriers to fully understand this. In terms of the cell biology concerning the localization of HCN1 versus HCN4, we have clarified in the discussion that it cannot be simply explained by palmitoylation of HCN channels or interaction with caveolins as has been explored in previous research. It remains an open question that is outside of the scope of this paper.

On the other hand, our paper presents a conceptual advance in the field of ion channel physiology. This is the first demonstration that the voltage sensor of an ion channel is directly regulated by membrane lipid compartmentalization, an emerging area in membrane biophysics. In many cases, an allosteric effect caused by changes in domains outside of the voltage-sensing domain can alter the voltage-dependent activation. Ionic current or gating current measurements are indirect methods to study the voltage sensor, and in our case, this type of demonstration necessitates the FRET methods used in this paper. Moreover, methods like cryo-EM/ET are still not successful in solving high-resolution structures in native cell membranes.

Major concerns:

1. The authors assert that “HCN4 channels exhibit a greater affinity for ordered domains compared to HCN1 channels.” I am not sure I understand this. The spectral FRET data shows that there is a similar FRET relationship between HCN4 and L10 probe (marking Lo) and S15 probe (Ld), while HCN1 has a high FRET relationship with S15 but not L10. To me, this seems to suggest that HCN4 is somewhat non-specific in preferring ordered and disordered domains, while HCN1 is preferentially excluded from ordered domains. Also it is worth noting that there is no real measurement of affinity here. Furthermore, how does one exclude the possibility that HCN1 may have a different conformation in L10 domains that led to different amounts of FRET?

The term “affinity” has been removed, instead we wrote, HCN4 exhibits a greater propensity for incorporation into ordered lipid domains compared to HCN1. Since we used an overexpression system, we expected that some or a significant amount of HCN4 would be localized in the S15 dominant lipid domains. The FRET efficiency is related to the expression level of donors and acceptors. The focus here is that HCN4 has a much higher likelihood to localize in the ordered domains than HCN1 does, and thus more vulnerable to disruption of the ordered lipid domains.

We recognized the potential distinctions arising from varying structural configurations of HCN1 and HCN4 channels within distinct lipid domains, as assessed by L10 and S15. Consequently, we designed an alternative approach (depicted in Figure 2) employing dyes conjugated to cholera toxin subunit B, along with noncanonical amino acids introduced into the highly conserved voltage-sensing domain of HCN channels. The outcomes presented parallel findings wherein HCN4 exhibited a notably greater FRET signal with the cholera toxin subunit B probe, as opposed to comparable experiments involving HCN1. This observation suggests that the L10 and S15 FRET experiments being substantially influenced by dissimilar channel conformations are less likely.

2. For a substantial number of figures, conclusions are drawn from quite small sample sizes, which I assume here refer to cells(?). In some cases, it is only $n = 3$. Is there sufficient power to draw statistical inferences with such small sample sizes?

We have done additional experiments to increase the n number to be at least 4 cells, and in most cases $n > 5$. Conclusions drawn are based on appropriate statistical analysis. In addition, a notable strength of lifetime-based fluorescence measurements, when contrasted with conventional intensity-based fluorescence measurements, resides in its enhanced precision.

3. In Fig. 3c, the authors show that a change in voltage also changes L-anap lifetime. However, if ordered domains are disrupted, this change vanishes. These data are then used to suggest that voltage sensor rearrangement in HCN4 may differ when membrane lipid domain is disrupted. The dataset is quite weak in my opinion. As noted in point 3, the sample size is quite limited. Furthermore, the axis limits are adjusted to artificially accentuate the changes. In the end though, L-anap elicits a very mild change in lifetime (<0.1 ns or ~3% change in signal). If you look closely, there is a reduction in lifetime of L-anap even in the presence of beta-cyclodextran. It is less prominent. Also, it may be helpful if the authors also performed similar experiments with HCN1. Presumably since it does not localize to ordered domains, the change in lifetime should be minimal.

We agree with the reviewer that a deviation of 0.1 ns might appear small but after repeating the experiments to expand the sample size, we confirmed that it's reproducible and significantly different from the control. Particularly, lifetime-based solvatochromic probes often report a small lifetime change. Furthermore, this small change in lifetime could be partially caused by using the membrane-permeable methyl-ester form of L-Anap that was applied for its incorporation, it is noteworthy that L-Anap might have exhibited nonspecific incorporation into the membrane. This nonspecific signal could potentially contribute to an underestimation of the observed change in fluorescence. It's similar to the traditional voltage-clamp fluorometry experiments where frequently the voltage-induced $\Delta F/F$ is < 5%. We have added this explanation in the Methods. Therefore, recalibrating the axis becomes essential in this particular context. Here, the key contrast lies in the fact that β -CD presence mitigated the high KCl-induced decrease in Anap lifetime.

Nonetheless, relying solely on these lifetime-based measurements does not yield a substantial amount of structural insight. This limitation prompted us to implement FRET experiments for a more quantitative understanding. In addition, the Alexa Fluor fluorophores used for FRET experiments are not membrane permeable.

At the request of the reviewer, we conducted additional experiments on HCN1, the results of which are now included in Figure 3., the high KCl-induced decrease in Anap lifetime did not exhibit statistical significance ($n = 5-6$). As the reviewer predicted, the result is different from the control condition for HCN4 but similar to when HCN4 was treated with β -CD. This is consistent with that HCN1 does not localize to ordered lipid domains.

4. Fig. 5g axis limits should be adjusted such that the minimum is 0 or at least a common value between resting and 120 KCl. There are other figures with the same issue.

We have adjusted the axis in Fig 5g to start from the same value between resting and 120 mM KCl.

5. "These findings imply that the conformation of the voltage-sensing domain (VSD) in HCN4 channels is influenced by membrane lipid domains. In contrast, HCN1 channels, predominantly found in disordered domains, appear to be less susceptible to changes induced by β -CD treatment" If this is the case, then, why is it that, disrupting the ordered domain in Fig. 5g show a strong reduction in FLIM-

FRET efficiency similar to the change observed with HCN4 (albeit in reverse). One would've expected there to be no effect of beta-cyclodextran here.

In Fig. 5, as the reviewer pointed out, there is a change in FRET in a reversed direction for the HCN1 L254/T171 pair when using transition metal FRET (tmFRET). This is likely due to the significantly smaller R_0 value of tmFRET, rendering it more attuned to even minor alterations in distance. This observation aligns with the notion that the response of the voltage sensor to β -CD treatment diverges between HCN1 and HCN4, potentially exhibiting variations not detectable through conventional sensitized-emission FRET, as depicted in Figure 4. Moreover, the dissimilar labeling strategies involving picolyl-azide AF555 and azido-DOTA could contribute to this difference. The relatively shorter linker length of azido-DOTA enables a better depiction of protein backbone movements. Furthermore, the β -CD treatment might prompt slight rearrangements in the acceptor site. Overall, the relevant tmFRET change is less pronounced for HCN1 than for HCN4 in the resting state, and no statistical significance for the I262/T171 pair, indicating that the β -CD effect is still more significant for the HCN4 than for the HCN1. We have provided additional explanation and discussion in the text.

6. Also in Fig. 4h, with HCN4, the FLIM-FRET efficiency decreases upon β -CD. However, the change seems to opposite with tmFRET. Is there an explanation for this? I understand that tmFRET is more sensitive to within-subunit changes. Does this mean the intrasubunit probes get closer but the intersubunit FRET gets further? If this is the case, could you make a plausible case for what the intra versus inter subunit conformational change is?

Please refer to the response to the previous comment #5. In addition, we do not think the tmFRET change in the Fig 5 is caused by intersubunit FRET since the R_0 is considerably short and much shorter than the corresponding distances between subunits. We have clarified this in the paper by explicitly listing the inter-subunit distances of these FRET pairs in the main text.

7. Is membrane expression of HCN1 and HCN4 strongly reduced upon incorporating ncAAs following dual codon-suppression? Based on exemplar traces in Supplementary Fig. 1 versus 5, the amplitudes are 50-fold lower? Is this because the chemical modification affects channel trafficking or just reconstitution is low efficacy. Also, how does this affect FLIM-FRET measurement? For instance, voltage depolarization or hyperpolarization may only affect a subset of channels that are at the plasma membrane and there could be substantial background signal from near membrane channels in vesicles?

The use of dual codon suppression introduces a challenge in the breaking-in procedure to establish the whole-cell configuration for patch clamp. We think this difficulty is primarily attributed to the incorporation of the ochre stop codon, which seems to negatively impact cellular health. This predicament was anticipated due to the expansion from single amber stop-codon suppression to the dual stop-codon nonsense suppression, and we have delved into this matter within the discussion section. To address this challenge, the patch-clamp experiments performed subsequent to the dual codon suppression, adopted the cell-attached configuration. In this configuration, currents were recorded from a portion of the plasma membrane enclosed within the patch electrode. Consequently,

the observed current amplitude appears notably reduced when contrasted with measurements conducted using the whole-cell configuration.

We performed additional “whole-cell versus cell-attached” patch-clamp experiments using the same tsA cell transfected with wild-type HCN4-YFP constructs, the results of which have been added to the Fig. S5 e and f (using the same voltage protocol as the rest of the panels). Despite that the current amplitude is much smaller using the cell-attached configuration, the G-V relationship is mostly the same comparing these two configurations.

Phasor plot FLIM has the advantage of distinctly differentiating between signals originating from the background outside of the membrane and those within the plasma membrane. Consequently, the process of excluding background interference outside of the membrane becomes significantly more straightforward compared to conducting measurements reliant on fluorescence intensity. Further elucidation on this matter can be found in the Method section.

8. Does beta-CD affect HCN1 versus HCN4 currents? If there is a change in VSD movement, one would envision there should be an appreciable change in channel biophysical properties for HCN4 channels but HCN1 should presumably be resistant to these? It may be helpful to determine the magnitude of changes with beta-CD in the specific expression system they are using since the amount and nature of lipid domains could vary depending on the expression system.

Previous research has done this type of experiments on human HCN1 and human HCN4 in the same overexpression system using HEK cells (cited and highlighted in the introduction and results). β -CD treatment does not affect the biophysical properties of HCN1 but significantly shifted the G-V relationship of HCN4. To show the functional effect of β -CD in our specific system, we reproduced the previous results, and have added the data to Fig. S1 f, g.

Reviewers' Comments:

Reviewer #1:

Remarks to the Author:

All my concerns have been addressed.

Reviewer #2:

Remarks to the Author:

The authors have adequately addressed all my concerns. My only comment is to add the exact statistical test used in the respective figure legend (i.e. which one is T-test, which one is ANOVA/Tukey's test).